# The gut microbiota-bile acid axis links the positive association between chronic insomnia and cardiometabolic diseases

Zengliang Jiang[1,7], Lai-bao Zhuo[2,7], Yan He[3,4,7], Yuanqing Fu [1,5,6], Luqi Shen[1,5,6], Fengzhe Xu[1,5], Wanglong Gou[1,5], Zelei Miao[1,5], Menglei Shuai[1,5], Yuhui Liang[1,5], Congmei Xiao[1,5], Xinxiu Liang[1,5], Yunyi Tian[1,5], Jiali Wang[1,5], Jun Tang[1,5,6], Kui Deng[1,5,6], Hongwei Zhou[3,4✉], Yu-ming Chen [2✉] & Ju-Sheng Zheng[1,5,6✉]

Evidence from human cohorts indicates that chronic insomnia is associated with higher risk of cardiometabolic diseases (CMD), yet whether gut microbiota plays a role is unclear. Here, in a longitudinal cohort ($n = 1809$), we find that the gut microbiota-bile acid axis may link the positive association between chronic insomnia and CMD. *Ruminococcaceae UCG-002* and *Ruminococcaceae UCG-003* are the main genera mediating the positive association between chronic insomnia and CMD. These results are also observed in an independent cross-sectional cohort ($n = 6122$). The inverse associations between those gut microbial bio-markers and CMD are mediated by certain bile acids (isolithocholic acid, muro cholic acid and nor cholic acid). Habitual tea consumption is prospectively associated with the identified gut microbiota and bile acids in an opposite direction compared with chronic insomnia. Our work suggests that microbiota-bile acid axis may be a potential intervention target for reducing the impact of chronic insomnia on cardiometabolic health.

[1] Key Laboratory of Growth Regulation and Translational Research of Zhejiang Province, School of Life Sciences, Westlake University, Hangzhou, China. [2] Guangdong Provincial Key Laboratory of Food, Nutrition and Health; Department of Epidemiology, School of Public Health, Sun Yat-sen University, Guangzhou, China. [3] Microbiome Medicine Center, Division of Laboratory Medicine, Zhujiang Hospital, Southern Medical University, Guangzhou, China. [4] State Key Laboratory of Organ Failure Research, Southern Medical University, Guangzhou, China. [5] Westlake Intelligent Biomarker Discovery Lab, Westlake Laboratory of Life Sciences and Biomedicine, Hangzhou, China. [6] Institute of Basic Medical Sciences, Westlake Institute for Advanced Study, Hangzhou, China. [7] These authors contributed equally: Zengliang Jiang, Lai-bao Zhuo, Yan He. ✉email: hzhou@smu.edu.cn; chenyum@mail.sysu.edu.cn; zhengjusheng@westlake.edu.cn

Chronic insomnia is a common sleep disorder with a current estimated global prevalence rate of ~10–20% [1–3]. Features of chronic insomnia include difficulty falling asleep, difficulty maintaining sleep, and awakening in the early morning, together with daytime fatigue, attention deficits and mood instability[4]. In the past decade, numerous observational studies have indicated that chronic insomnia is associated with higher risk of cardiometabolic diseases (CMD), such as type 2 diabetes (T2D) and cardiovascular diseases[5–9]. However, the mechanism that underlies the association between chronic insomnia and CMD has yet to be identified, so novel cost-effective therapeutic strategies have yet to be developed.

The gut microbiota is vital to human health[10,11]. Notably, the brain-gut axis has been intensively studied in the past few years[12–14]. Prior studies have reported that the gut microbiota exhibits circadian rhythms, which interact with host circadian rhythms[15–17]. Sleep disturbances, such as chronic insomnia, can in turn disrupt microbial circadian rhythms, thus influencing gut microbial composition and function[18–22]. On the other hand, gut microbial dysbiosis is associated with the development of CMD, and has a substantial impact on the metabolic health[23–29]. In addition, the dysregulation of bile acid metabolism and its interaction with the gut microbiota are also closely associated with host metabolic health[30–33]. Repeated sleep disruption in mice has led to a persistent change in gut microbiota composition and changes in bile acid metabolism[34–36]. Therefore, we hypothesize that the gut microbiota-bile acid axis may play a role in linking chronic insomnia and CMD. Notably, evidence from large-scale human cohorts is sparse.

In the present study, we examined the longitudinal association of chronic insomnia status over ~6 years with gut microbiota and bile acid profiles in a Chinese prospective cohort study, including 1809 participants from the Guangzhou Nutrition and Health Study (GNHS)[37]. We further investigated whether these altered gut microbiota or bile acids could mediate the chronic insomnia-CMD association in the GNHS. Validation of the above associations was conducted in an independent large cross-sectional study involving 6122 participants from the Guangdong Gut Microbiome Project (GGMP)[38].

## Results

**Large gut microbiome cohorts with deep phenotyping data.** The present study was based on the GNHS, a community-based prospective cohort including 4048 participants of Han Chinese ethnicity, who were recruited between 2008 and 2013. Validation of the results from GNHS was based on data from the GGMP, a large community-based cross-sectional cohort conducted between 2015 and 2016 including 7,009 participants with high quality gut microbiome data. We profiled the gut microbiota (16S rRNA sequencing) in the GNHS ($n = 1809$) and GGMP ($n = 6122$) cohort participants, who provided detailed information on chronic insomnia status, CMD and related risk factors (Fig. 1a, and Tables 1 and 2). We also performed targeted fecal bile acid metabolome analyses among 954 participants from the GNHS cohort (Fig. 1a). In the GNHS, the participants were divided into four groups according to their chronic insomnia status over a period of 6 years prior to the collection of stool samples: (i) Long-term healthy group (i.e., without chronic insomnia at baseline or follow-up), (ii) Recovery group (i.e., from chronic insomnia at baseline to normal at follow-up), (iii) New-onset group (i.e., without chronic insomnia at baseline but with chronic insomnia at follow-up), and (iv) Long-term chronic insomnia group (i.e., with chronic insomnia at baseline and follow-up) (Fig. 1a and Methods). For the GGMP, given the cross-sectional study design, the participants were divided into two groups: (i) Non-chronic insomnia group, and (ii) Chronic insomnia group (Methods).

**Chronic insomnia was associated with the diversity of the gut microbiota.** We first investigated the longitudinal association of chronic insomnia status of the GNHS participants with microbioal α-/β- diversity. We found that there were significant differences in the microbial β- diversity of the New-onset group and Long-term chronic insomnia group, compared to the Long-term healthy group (Fig. 1b). The α-diversity parameters (Observed species, Chao 1 index and ACE index) of the New-onset group and Long-term chronic insomnia group were significantly lower than those of the Long-term healthy group (Fig. 1b and Supplementary Fig. 1). The α-diversity parameter the Shannon index of the Long-term chronic insomnia group was significantly lower than that of the Long-term healthy group (Supplementary Fig. 1). The α-diversity parameter the Simpson index was not significantly different among the four chronic insomnia status groups (Supplementary Fig. 1). Consistent results were observed by using three different statistical models (Fig. 1b and Supplementary Fig. 1), suggesting that chronic insomnia was robustly associated with the gut microbiota structure.

We combined the New-onset group and Long-term chronic insomnia group into one Chronic insomnia group to increase the sample size, given that there was no significant difference in the microbiota structure between the two groups. Chronic insomnia in the GNHS participants was associated with lower levels of Observed species ($p < 0.01$), Shannon index ($p < 0.05$), Chao 1 index ($p < 0.001$) and ACE index ($p < 0.001$), respectively (Fig. 2a and Supplementary Fig. 2). Principal coordinate analysis of the gut microbial profiles in the GNHS showed a significant shift in the gut microbiota composition of the Chronic insomnia group compared to the Long-term healthy group ($p < 0.01$; PERMANOVA test with 999 permutations) (Fig. 2b). A similar pattern was observed for the shift in the structure of the gut microbiota in the GGMP (Fig. 2a, b and Supplementary Fig. 3).

**Chronic insomnia was associated with specific gut microbes and bile acids.** We used Multivariate Analysis by Linear Models (MaAsLin) adjusted for potential confounders to identify potential microbial biomarkers (as outcome variables) of chronic insomnia (as a predictor in the model). *Ruminococcaceae UCG-002* and *Ruminococcaceae UCG-003* were identified as the microbial biomarkers of chronic insomnia (Fig. 2c and Supplementary Table 1). In the GNHS, Chronic insomnia group was associated with lower levels of *Ruminococcaceae UCG-002* ($\beta$: −0.19, 95% CI: −0.32 to −0.06) and *Ruminococcaceae UCG-003* ($\beta$: −0.20, 95% CI: −0.33 to −0.07), compared with the Long-term healthy group (Fig. 2d). These results were also observed in the GGMP (Fig. 2d). The results of the sensitivity analysis suggested that different covariate adjustments did not substantially affect the results in the GGMP (Supplementary Fig. 4). In addition, chronic insomnia had no interactions with age or sex for *Ruminococcaceae UCG-002* and *Ruminococcaceae UCG-003* (Supplementary Table 2). To assess the potential influence of the number of insomnia symptoms, we conducted a secondary analysis of the GGMP participants, for whom the data were available, and found that the per unit change in the chronic insomnia symptom score was inversely associated with a per 1-SD change in *Ruminococcaceae UCG-002* ($\beta$: −0.04, 95% CI: −0.06 to −0.02, $p < 0.001$) and *Ruminococcaceae UCG-003* ($\beta$: −0.04, 95% CI: −0.07 to −0.01, $p = 0.002$) (Supplementary Table 3).

We next used orthogonal partial least squares discrimination analysis (OPLS-DA) to identify potential fecal bile acids associated with chronic insomnia (Supplementary Fig. 5) and then used linear regression to confirm the chronic insomnia-bile acid associations in the GNHS (Fig. 2e). Chronic insomnia was associated with higher levels of muro cholic acid (MCA, $p = 0.046$) and nor cholic acid (NorCA, $p = 0.046$) and with

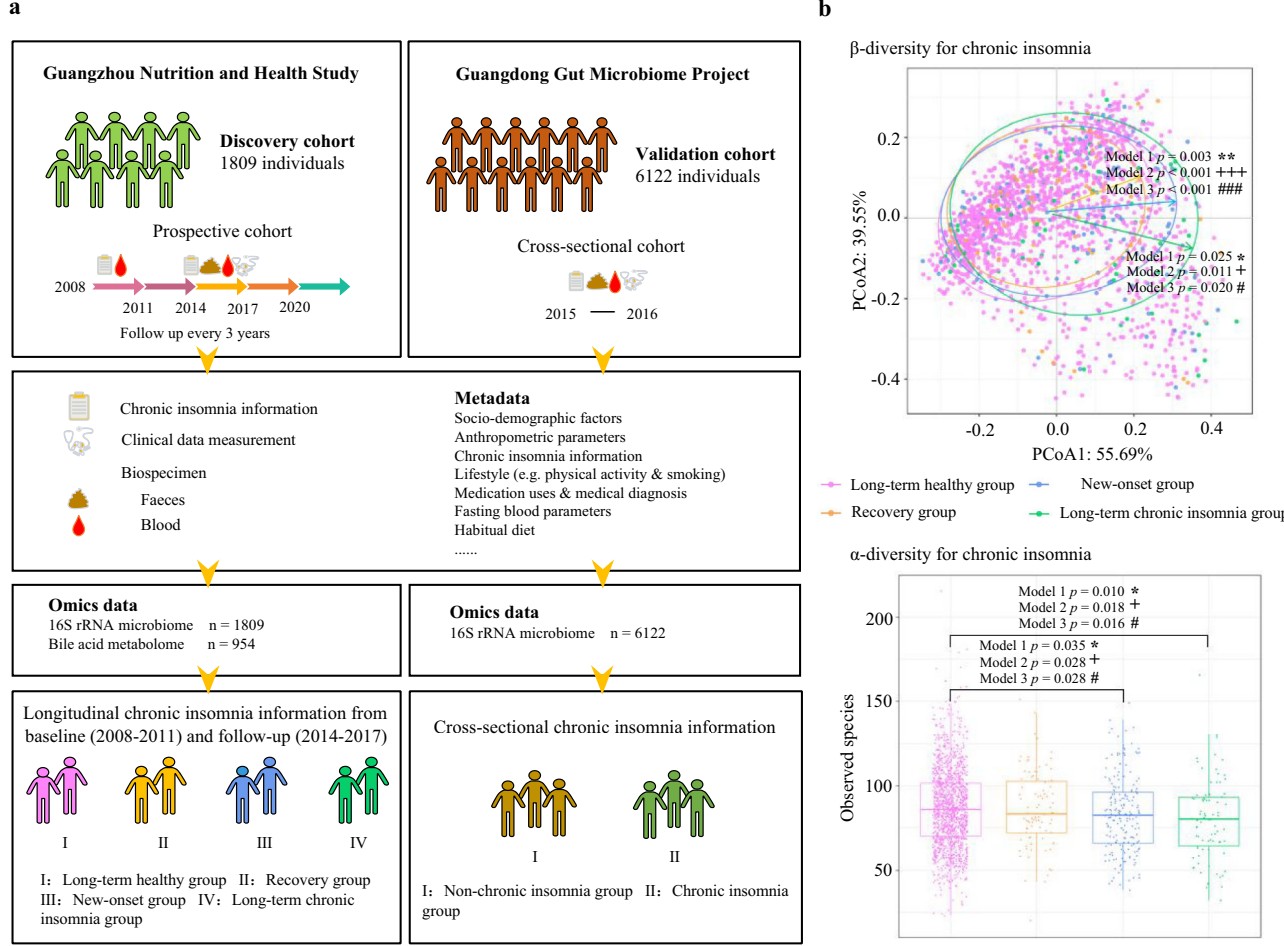

**Fig. 1 Study diagram and gut microbiota diversity by chronic insomnia status. a** Conceptual diagram of the present study. **b** The association of chronic insomnia with α-/β- microbial diversity among the four groups (n = 1809). The association of chronic insomnia with the overall microbial α-diversity parameter Observed species was evaluated using a multivariable linear regression, adjusted for potential confounding factors (three models in the text). Box plots indicate median and interquartile range (IQR). The upper and lower whiskers indicate 1.5 times the IQR from above the upper quartile and below the lower quartile. The results of Shannon index, Chao 1 index, ACE index and Simpson index are reported in Supplementary Fig. 1. β-diversity was evaluated using principal coordinate analysis (PCoA) plot based on Bray-Cutis distance at the genus level. Permutational ANOVA (PERMANOVA) (999 permutations) was used to identify the variation of β-diversity in human gut microbiota structure comparing the four groups, adjusted for the same covariates. The Benjamini-Hochberg method was used to adjust p values for multiple testing. Value with symbol is significantly different (model 1: *p < 0.05, **p < 0.01, ***p < 0.001; model 2: +p < 0.05, ++p < 0.01, +++p < 0.001; model 3: #p < 0.05, ##p < 0.01, ###p < 0.001). All statistical tests were two-sided. Source data are provided as a Source Data file.

lower levels of isolithocholic acid (IsoLCA, p = 0.029), lithocholic acid (LCA, p = 0.035) and ursodeoxycholic acid (UDCA, p = 0.039) (Fig. 2e). The results of the sensitivity analysis showed that adding dietary cholesterol intake and fiber intake as additional covariates did not substantially affect the association of chronic insomnia with bile acids (Supplementary Table 4). Co-occurrence network analysis based on the partial correlation coefficient showed that *Ruminococcaceae UCG-002* and *Ruminococcaceae UCG-003* were positively associated with secondary bile acids (IsoLCA, LCA, and UDCA), and inversely associated with primary bile acids (MCA and NorCA) (p < 0.001) (Fig. 2f). In addition, chronic insomnia was not associated with short-chain fatty acids, aromatic amino acids, or their derivatives (Supplementary Table 5). These results indicated that chronic insomnia might have a significant impact on the gut microbiota-bile acid axis.

**Chronic insomnia-related gut microbial features and bile acids were associated with CMD and risk factors.** To further investigate whether the chronic insomnia-related gut microbiota or

bile acids play a role in CMD, we used multivariable logistic regression to examine the association of the chronic insomnia-related gut microbiota or bile acids with CMD. In the cross-sectional analysis of the GNHS participants, a per standard deviation (SD)-unit increment in *Ruminococcaceae UCG-002* was associated with a 24% lower risk of metabolic syndrome (MetS) (OR: 0.76, 95% CI: 0.66–0.86), a 22% lower risk of T2D (OR: 0.78, 95% CI: 0.67–0.91), and 13% lower risk of dyslipidemia (OR: 0.87, 95% CI: 0.79–0.97) (Fig. 3a). Each per SD-unit increment in *Ruminococcaceae UCG-003* was associated with a 23% lower risk of MetS (OR: 0.77, 95% CI: 0.67–0.88) (Fig. 3b). IsoLCA was inversely associated with T2D (OR: 0.74, 95% CI: 0.61–0.90) (Fig. 3c). MCA and NorCA were positively associated with MetS (OR: 1.33, 95% CI: 1.13–1.58; OR: 1.36, 95% CI: 1.15–1.61), respectively (Fig. 3c).

The majority of the results from the GNHS could be also observed in the GGMP. Meta-analysis of results from the two cohorts consistently showed that *Ruminococcaceae UCG-002* and *Ruminococcaceae UCG-003* were significantly inversely associated with MetS (Pooled OR: 0.82, 95% CI: 0.72–0.93; Pooled OR: 0.82,

**Table 1 Characteristics of the study participants in the Guangzhou Nutrition and Health Study[a].**

| Characteristics | Total | Groups | | | | |
| --- | --- | --- | --- | --- | --- | --- |
| | | Long-term healthy | Recovery | New-onset | Long-term chronic insomnia | *p* value |
| *n* | 1809 | 1443 | 90 | 198 | 78 | |
| Age, y | 58.5 (6.1) | 58.5 (6.1) | 58.2 (6.2) | 58.1 (5.7) | 59.6 (5.8) | 0.33 |
| Sex, *n* (% of women) | 1219 (67.4) | 927 (64.2) | 71 (78.9) | 155 (78.3) | 66 (84.6) | <0.001 |
| BMI, kg/m$^2$ | 23.2 (3.0) | 23.4 (3.0) | 22.8 (3.4) | 22.9 (3.0) | 22.2 (2.8) | 0.002 |
| Total energy intake, kcal/d | 1748 (489) | 1738 (546) | 1724 (477) | 1638 (374) | 1748 (489) | 0.260 |
| Physical activity, MET h/d | 40.6 (13.9) | 41.8 (15.1) | 40.8 (15.1) | 40.5 (14.2) | 40.6 (13.9) | 0.870 |
| Vegetable intake, g/d | 370 (177) | 336 (162) | 381 (161) | 380 (183) | 370 (177) | 0.220 |
| Fruit intake, g/d | 146 (109) | 148 (111) | 134 (91) | 139 (96) | 142 (135) | 0.490 |
| Red and processed meat intake, g/d | 82 (52) | 83 (53) | 87 (52) | 78 (43.0) | 78 (46) | 0.430 |
| Fish intake, g/d | 50 (52) | 50.8 (55.0) | 51.3 (38.4) | 47.5 (33.7) | 43.9 (33.3) | 0.580 |
| Dairy products intake, g/d | 17.2 (14.4) | 17.3 (14.6) | 17.1 (12.6) | 16.6 (14.7) | 17.0 (12.0) | 0.940 |
| Coffee intake, g/d | 8.5 (34.1) | 8.5 (34.5) | 7.0 (20.4) | 9.5 (31.4) | 7.1 (45.5) | 0.920 |
| Current tea drinker, *n* (%) | 967 (53.6) | 791 (54.8) | 39 (43.3) | 98 (49.5) | 39 (50.0) | 0.094 |
| Current alcohol drinker, *n* (%) | 129 (7.1) | 106 (7.3) | 5 (5.6) | 16 (8.1) | 2 (2.6) | 0.370 |
| Current smoker, *n* (%) | 282 (15.6) | 245 (17.0) | 10 (11.1) | 20 (10.1) | 7 (9.0) | 0.014 |
| Income level, *n* (%) | | | | | | 0.350 |
| ≤500 ¥/mo | 26 (1.4) | 19 (1.3) | 1 (1.1) | 4 (2.0) | 2 (2.6) | |
| 501–1500 ¥/mo | 391 (21.6) | 301 (20.9) | 26 (28.9) | 49 (24.7) | 15 (19.2) | |
| 1501–3000 ¥/mo | 1156 (63.9) | 930 (64.4) | 55 (61.1) | 116 (58.6) | 55 (70.5) | |
| >3000 ¥/mo | 236 (13.0) | 193 (13.4) | 8 (8.9) | 29 (14.6) | 6 (7.7) | |
| Education, *n* (%) | | | | | | 0.400 |
| Middle school or lower | 395 (27.4) | 19 (21.1) | 51 (25.8) | 27 (34.6) | 395 (27.4) | |
| High school or professional college | 664 (46.0) | 40 (44.4) | 92 (46.5) | 35 (44.9) | 664 (46.0) | |
| University | 384 (26.6) | 31 (34.4) | 55 (27.8) | 16 (20.5) | 384 (26.6) | |
| SBP | 121 (17) | 122 (17) | 118 (15) | 119 (16) | 116 (16) | 0.005 |
| DBP | 74 (12) | 74 (13) | 73 (10) | 74 (10) | 72 (9) | 0.540 |
| TG | 1.6 (1.3) | 1.6 (1.2) | 1.4 (0.6) | 1.5 (0.7) | 1.6 (0.9) | 0.350 |
| TC | 5.5 (1.1) | 5.5 (1.1) | 5.3 (1.1) | 5.6 (1.1) | 5.7 (1.2) | 0.031 |
| LDL | 3.6 (1.0) | 3.6 (1.0) | 3.5 (1.0) | 3.7 (1.0) | 3.8 (1.1) | 0.220 |
| HDL | 1.5 (0.4) | 1.5 (0.4) | 1.5 (0.4) | 1.5 (0.4) | 1.5 (0.4) | 0.370 |
| Glucose, mmol/L | 5.5 (1.3) | 5.5 (1.3) | 5.4 (1.0) | 5.4 (1.4) | 5.4 (1.4) | 0.500 |
| Insulin, μU/mL | 9.1 (6.6) | 9.2 (6.9) | 8.6 (4.3) | 8.4 (5.1) | 8.8 (4.6) | 0.510 |
| HbA1c, % | 5.8 (0.8) | 5.8 (0.8) | 5.8 (0.8) | 5.8 (0.9) | 5.8 (0.6) | 0.810 |
| Medication use, *n* (%) | | | | | | |
| Hypertension | 96 (5.3) | 45 (3.1) | 2 (2.2) | 4 (2.0) | 5 (6.4) | 0.280 |
| Hyperlipidaemia | 108 (6.0) | 91 (6.3) | 4 (4.4) | 10 (5.1) | 3 (3.8) | 0.660 |
| T2D | 56 (3.1) | 45 (3.1) | 2 (2.2) | 4 (2.0) | 5 (6.4) | 0.280 |

*HbA1c* glycated hemoglobin, *T2D* type 2 diabetes, *SBP* systolic blood pressure, *DBP* diastolic blood pressure, *TG* triglycerides, *TC* total cholesterol, *HDL* high-density lipoprotein cholesterol, *LDL* low-density lipoprotein cholesterol.
[a]Data are expressed as mean with standard deviation (SD) for continuous variables and *n* (%) for categorical variables; *p* value represents the comparison among groups using analysis of variance (ANOVA) or Pearson's chi-squared; All statistical tests were two-sided.

95% CI: 0.77–0.88), T2D (Pooled OR: 0.84, 95% CI: 0.75–0.95; Pooled OR: 0.87, 95% CI: 0.81–0.95), dyslipidemia (Pooled OR: 0.91, 95% CI: 0.87–0.95; Pooled OR: 0.88, 95% CI: 0.84–0.92) and coronary heart disease (CHD) (Pooled OR: 0.88, 95% CI: 0.82–0.94; Pooled OR: 0.93, 95% CI: 0.87–0.99), respectively (Fig. 3a, b). Consistent results were observed in the sensitivity analyses (Supplementary Fig. 6). *Ruminococcaceae UCG-002* was interacted with sex in the risk of dyslipidaemia ($p_{interaction}$ = 0.003) (Supplementary Table 6). The stratified analyses by sex showed that the inverse association of *Ruminococcaceae UCG-002* with dyslipidaemia was significant among the men participants (Pooled OR: 0.87, 95% CI: 0.81–0.93, $p < 0.001$), but not among the women participants (Pooled OR: 0.96, 95% CI: 0.88–1.05, $p = 0.394$) (Supplementary Table 6).

In the longitudinal analysis among GNHS, we found that *Ruminococcaceae UCG-002* and *Ruminococcaceae UCG-003* were inversely associated with the incidence of dyslipidemia (OR: 0.72, 95% CI: 0.57–0.90; OR: 0.74, 95% CI: 0.60–0.91), respectively

(Fig. 3d). In addition, *Ruminococcaceae UCG-002* was significantly inversely associated with BMI (β: −0.28, 95% CI: −0.43 to −0.12) (Fig. 3e).

To evaluate whether bile acids can mediate the relationship between the gut microbiota and CMD (Fig. 3f), we applied mediation analysis, which showed that the inverse association of the chronic insomnia-related gut microbial biomarkers with MetS and T2D were mediated by some specific bile acids (Fig. 3g, h). MCA mediated the association of *Ruminococcaceae UCG-002* and *Ruminococcaceae UCG-003* with the risk of MetS (50.8%, $p_{mediation} < 0.001$; 39.5%, $p_{mediation} < 0.001$, respectively, Fig. 3g, h). NorCA mediated the association of *Ruminococcaceae UCG-002* and *Ruminococcaceae UCG-003* with the MetS risk (32.9%, $p_{mediation} < 0.001$; 31.9%, $p_{mediation} < 0.001$, respectively, Fig. 3g, h). In addition, IsoLCA mediated the association of *Ruminococcaceae UCG-002* and *Ruminococcaceae UCG-003* with the risk of T2D (41.7%, $p_{mediation} = 0.040$; 53.2%, $p_{mediation} < 0.001$, respectively, Fig. 3g, h). Sensitivity analysis for mediation effects indicated that the results of

**Table 2 Characteristics of study participants from the Guangdong Gut Microbiome Project[a].**

| Characteristics | Total population | Groups | | |
|---|---|---|---|---|
| | | Non-chronic insomnia | Chronic insomnia | *p* value |
| *n* | 6122 | 2963 | 3159 | |
| Age, y | 52.9 (14.7) | 49.8 (14.6) | 55.8 (14.2) | <0.001 |
| Sex, *n* (% of women) | 3399 (55.5) | 1505 (50.8) | 1894 (60.0) | <0.001 |
| BMI, kg/m² | 23.3 (3.5) | 23.4 (3.6) | 23.2 (3.5) | 0.011 |
| Vegetable intake, g/d | 335 (231) | 337 (216) | 333 (243) | 0.560 |
| Fruit intake, g/d | 80 (117) | 81 (111) | 78 (122) | 0.190 |
| Red and processed meat intake, g/d | 129 (121) | 134 (122) | 125 (119) | 0.007 |
| Current alcohol drinker, *n* (%) | 2384 (38.9) | 1182 (39.9) | 1202 (38.1) | 0.140 |
| Current smoker, *n* (%) | 1563 (25.5) | 851 (28.7) | 712 (22.5) | <0.001 |
| Education, *n* (%) | | | | <0.001 |
| Middle school or lower | 4649 (75.9) | 2134 (72.0) | 2515 (79.6) | |
| High school or professional college | 1215 (19.8) | 674 (22.7) | 541 (17.1) | |
| University | 258 (4.2) | 155 (5.2) | 103 (3.3) | |
| SBP | 132 (22) | 130 (21) | 134 (23) | <0.001 |
| DBP | 78 (12) | 78 (12) | 78 (12) | 0.370 |
| TG | 1.4 (1.5) | 1.4 (1.6) | 1.4 (1.3) | 0.640 |
| TC | 5.3 (0.9) | 5.2 (0.8) | 5.3 (0.9) | 0.002 |
| LDL | 3.3 (0.9) | 3.2 (0.9) | 3.3 (1.0) | <0.001 |
| HDL | 1.3 (0.5) | 1.3 (0.5) | 1.3 (0.5) | 0.021 |
| FBG | 5.6 (1.6) | 5.5 (1.6) | 5.7 (1.7) | <0.001 |

*SBP* systolic blood pressure, *DBP* diastolic blood pressure, *TG* triglycerides, *TC* total cholesterol, *HDL* high-density lipoprotein cholesterol, *LDL* low-density lipoprotein cholesterol, *FBG* Fasting blood glucose.
[a]Data are expressed as mean with standard deviation (SD) for continuous variables and *n* (%) for categorical variables; *p* value represents the comparison among groups using analysis of variance (ANOVA) or Pearson's chi-square test. All statistical tests were two-sided.

the above mediation analysis were relatively robust to the possible existence of an unmeasured confounder (Supplementary Table 7).

**Habitual dietary intakes and gut microbiota-bile acid axis.** We used multivariable linear regression models to investigate the longitudinal association of dietary factors with chronic insomnia-related microbial and bile acid biomarkers in the GNHS participants without chronic insomnia or CMD at baseline. Among different food groups, we found that only tea consumption was positively significantly associated with *Ruminococcaceae UCG-002* (β: 0.28, 95% CI: 0.13–0.43; *p* = 0.002), and inversely associated with bile acid NorCA (β: −0.22, 95% CI: −0.42 to −0.02; *p* = 0.029, which was positively associated with chronic insomnia) (Fig. 4a, b and Supplementary Fig. 7). The tea consumption-*Ruminococcaceae UCG-002* association was also observed in the GGMP (β: 0.27, 95% CI: 0.08–0.47; *p* = 0.002) (Fig. 4a). Furthermore, a stratified analysis by tea consumption (yes versus no) among the GNHS participants showed that the inverse association between *Ruminococcaceae UCG-002* and CMD risk factors (especially for T2D (OR: 0.73, 95% CI: 0.60–0.89, *p* = 0.002) and dyslipidemia (OR: 0.85, 95% CI: 0.74–0.98, *p* = 0.024)) were generally stronger among those with habitual tea consumption (Table 3). In addition, meta-analysis of the results of the tea consumption-chronic insomnia association from the two cohorts showed that tea consumption (yes versus no) was inversely associated with the risk of chronic insomnia (Pooled OR: 0.72, 95% CI: 0.55–0.95, *p* = 0.020) (Supplementary Table 8). The results indicated that habitual tea consumption was associated with the gut microbiota-bile acid axis, which may potentially underlie the association between chronic insomnia and CMD (Fig. 4c).

**Discussion**

In the present study, we demonstrated that chronic insomnia was significantly associated with the structure and composition of the gut microbiota and specific bile acids. The chronic insomnia inverse-related gut microbiota *Ruminococcaceae UCG-002* and *Ruminococcaceae UCG-003* were significantly inversely associated with CMD and related traits. The gut microbial features of chronic insomnia and their relationships with CMD traits were also observed in an independent cohort (GGMP). Furthermore, we found that chronic insomnia-related bile acids (MCA, NorCA, and IsoLCA) may mediate the association of the identified microbial features with CMD traits. Finally, habitual tea consumption was associated with higher levels of *Ruminococcaceae UCG-002* and lower levels of the bile acid NorCA, and the tea drinking-*Ruminococcaceae UCG-002* association was also observed in the GGMP.

Recent human studies have shown that sleep duration and rhythm are associated with variations in the gut microbiota[18,39]. However, to date, evidence from large cohort studies on the associations between chronic insomnia and the gut microbiota is particularly lacking. Our results from two large cohort studies provided timely evidence supporting that chronic insomnia was associated with variations in the gut microbiota. Specifically, *Ruminococcaceae UCG-002* and *Ruminococcaceae UCG-003* were identified as the two potential genera inversely associated with chronic insomnia, and those microbes may be associated with host glucose homeostasis and lipid metabolism[40,41]. In addition to the gut microbiota, our study provided new insight into the relationship between chronic insomnia and bile acids. Bile acids are recognized as potent signaling molecules that impact glucose and lipid homeostasis through activation of Farnesoid-X receptor (FXR)[42–44] and Takeda-G-protein-receptor-5 (TGR5)[45]. As alterations in the gut microbiota would alter many other metabolites, we performed a secondary analysis of the association of chronic insomnia with short-chain fatty acids, aromatic amino acids and their derivatives[46,47]. Although we did not find any significant association for these two classes of metabolites, we could not rule out the possibility that chronic insomnia may be associated with other metabolites, which needs further investigation.

As indicated in previous studies[48,49], the gut microbiota-bile acid axis is vital to human health. In the past decade, evidence

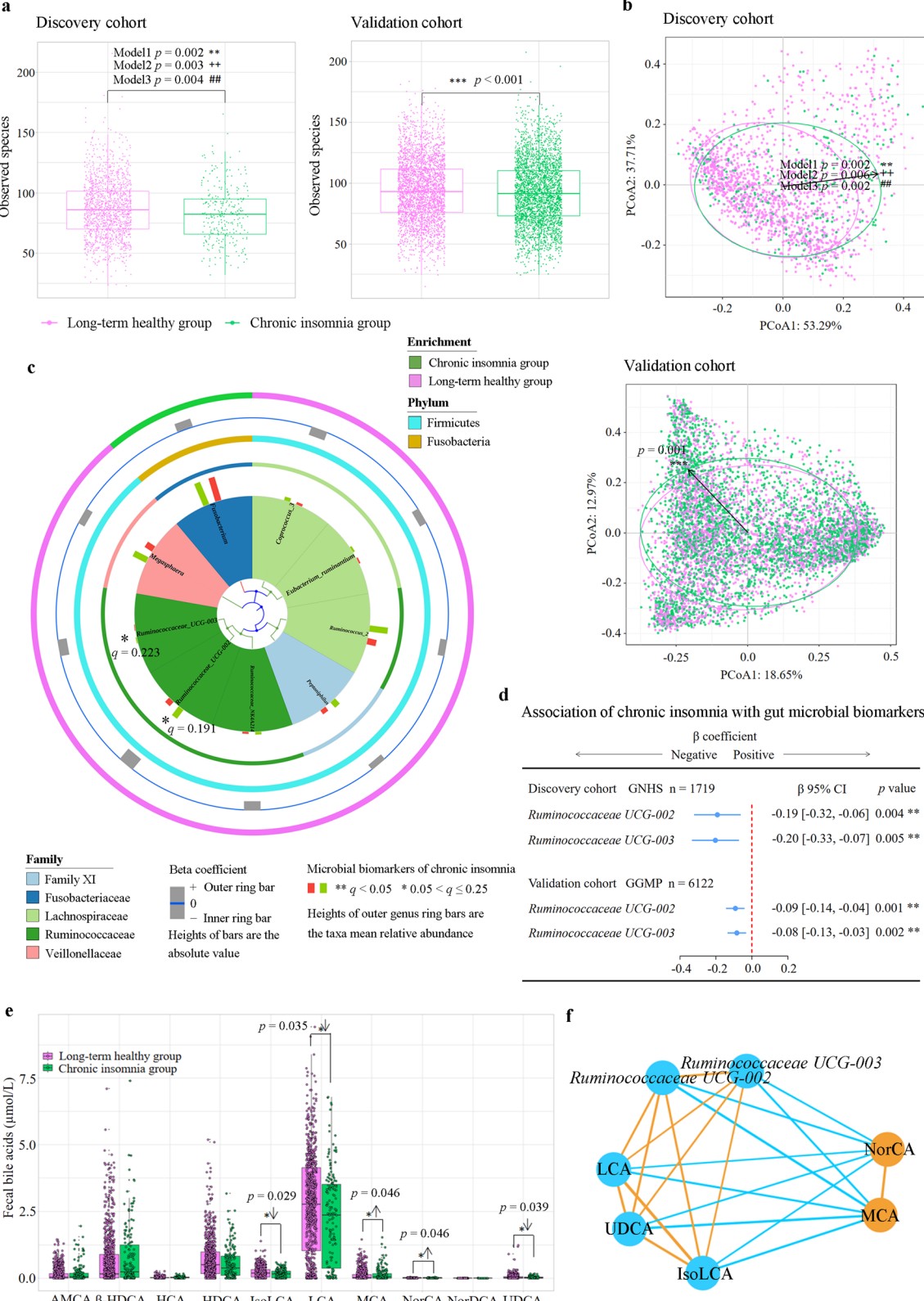

from human cohort studies has indicated that chronic insomnia is associated with higher risk of CMD[5–9,50,51], yet whether the gut microbiota-bile acid axis plays a role in the above association is unknown. Our study demonstrated that chronic insomnia-related gut microbial features were closely associated with CMD endpoints, which may be mediated by specific bile acids. The results were consistent with several recent studies[41,52–55]. One study

indicated that secondary bile acid metabolites (i.e., glycoursodeoxycholate) might link poor habitual sleep quality and coronary heart disease risk[52]. Another study showed that *Ruminococcaceae UCG-002* was positively associated with insulin sensitivity in patients with polycystic ovary syndrome[41]. In addition, another recent study demonstrated that *Ruminococcaceae UCG-002 and Ruminococcaceae UCG-003* were positively

**Fig. 2 Association of chronic insomnia with gut microbiota and bile acids. a** Observed species in the discovery cohort ($n = 1809$) and validation cohort ($n = 6122$). The results of Shannon index, Chao 1 index, ACE index and Simpson index are reported in Supplementary Fig. 2 (discovery cohort) and Supplementary Fig. 3 (validation cohort). $p$ value was calculated from multivariable-adjusted linear regression with three different models (Methods; model 1: *, model 2: +, model 3: #). Box plots indicate median and interquartile range (IQR). The upper and lower whiskers indicate 1.5 times the IQR from above the upper quartile and below the lower quartile. **b** β-diversity: principal coordinate analysis (PCoA) of genus-level Bray-Cutis distance in the discovery and validation cohorts. Permutational ANOVA (999 permutations) was used to identify the variation of β-diversity, adjusted for the same covariates as α-diversity. **c** Multivariate Analysis by Linear Models (MaAsLin) was used to identify the gut microbial biomarkers for chronic insomnia comparing Chronic insomnia group with Long-term healthy group. The $q$ values (false discovery rate adjusted $p$ value) were calculated using the Benjamini-Hochberg method (*$q < 0.25$, **$q < 0.05$). **d** Multivariable linear regression was used to assess the association of chronic insomnia with the gut microbial biomarkers in the discovery and validation cohorts, adjusted for the same covariates as α-diversity. Error bars are beta coefficient with 95% confidence intervals. **e**, Bile acid biomarkers of chronic insomnia in the GNHS ($n = 954$). Box plots indicate median and interquartile range. Orthogonal partial least squares discrimination analysis (OPLS-DA) (Supplementary Fig. 5) and multivariable-adjusted linear regression were used to identify potential bile acids associated with chronic insomnia (*$p < 0.05$, **$p < 0.01$, ***$p < 0.001$). **f** Partial correlation analysis was used to assess the interrelationships between the identified gut microbiota and bile acid biomarkers, adjusted for age, sex and BMI. Orange/sky blue circles indicate chronic insomnia-positive/negative biomarkers. Orange/sky blue lines indicate positive/negative associations. The Benjamini-Hochberg method was used to correct for multiple testing. All statistical tests were two-sided. Source data are provided as a Source Data file. α-MCA α-muricholic acid; β-HDCA β-hyodeoxycholic acid; HDCA hyodeoxycholic acid; HCA hyocholic acid; IsoLCA isolithocholic acid; LCA lithocholic acid; MCA Muro cholic acid; NorCA Nor cholic acid; NorDCA Nor deoxycholic acid; UDCA ursodeoxycholic acid.

associated with several plasma HDL subclasses, and were inversely associated with several plasma LDL subclasses, which had direct beneficial implications for cardiovascular health[53]. Several other studies have suggested that treatment with specific microbial derived secondary bile acids (obeticholic acid, deoxycholic acid, and glycodeoxycholic acid) in patients with T2D could improve insulin sensitivity and HbA1c levels[54,55].

The perturbations of the gut microbiota strongly affect bile acid metabolism, especially a failure to metabolize some primary bile acids leading to primary bile acid accumulation and secondary bile acid reduction[56,57]. *Ruminococcaceae UCG-002* and *Ruminococcaceae UCG-003* may have the ability to convert some primary bile acids into secondary bile acids as they belong to the bile salt hydrolase (BSH) and 7α-dehydroxylase-active family *Ruminococcaceae*, which harbors many secondary bile acid-producing genera such as *Faecalibacterium* and *Ruminniclostridium*[58,59]. Given the hormone-like functions of bile acids through activation of FXR and TGR5, a dysregulated bile acid pool can lead to perturbations in multiple pathological processes underlying CMD, such as immune regulation and lipid and glucose homeostasis[60,61].

We found a prospective association of habitual tea consumption with the identified gut microbiota and bile acids that was opposite to that of chronic insomnia. The major tea types consumed by the cohort participants were oolong tea, green tea, pu-erh tea, and black tea, which are beneficial to host metabolic health[62,63]. The mechanism underlying the association of habitual tea consumption with the gut microbiota-bile acid axis may be attributed to the rich content of tea polyphenols, flavonoids, alkaloids, and various antioxidant compounds, which are reported to modulate the gut microbiota composition and bile acid metabolism[64–66] and improve the circadian rhythm systems of the brain and gut[67,68]. Nevertheless, we cannot establish a causal relationship between tea consumption and CMD-related gut microbiota at this stage, and these above speculations should be considered with caution, especially given that caffeine in tea may exacerbate insomnia[69]. Randomized controlled trials are further needed to examine the effectiveness of habitual tea consumption on the gut microbiome.

The present study has several strengths. First, it is based on a large longitudinal cohort, given that evidence from the prospective relationship of chronic insomnia status with the gut microbiota is particularly lacking. Second, we used the gut microbiota-bile axis to interpret the connection between chronic insomnia and CMD, which provides novel mechanistic insight

into the above epidemiological association. Third, our main findings were also observed in another large cohort study. The present study also contains several limitations. First, this study is based on an observational study design, and residual confounders could not be avoided. Second, although we demonstrate that the gut microbiota-bile acid axis may link the association between chronic insomnia and CMD, the underlying causality remains unelucidated. Third, the replication cohort (GGMP) is a cross-sectional study, and the potential impact of the two slightly different definitions of chronic insomnia between the GNHS and GGMP is still unclear, although our results were also observed in the GGMP. Fourth, we did not collect information on sleep-disordered breathing, which is closely associated with chronic insomnia and may have an impact on the gut microbiota and bile acid metabolism[70–72]. Fifth, we conducted the mediation analysis for multiple bile acids using separate single mediator models; however, it is possible that these bile acids are highly correlated with each other or even have a causal association with each other, which needs further investigation. Finally, our two cohorts are both based on individuals of Chinese ethnicity, which may not be generalizable to other populations and ethnicities.

In summary, the present study indicates that chronic insomnia is associated with the structure and composition of the gut microbiota and specific bile acids. The gut microbiota-bile acid axis may play an essential role in linking chronic insomnia and CMD outcomes. Habitual tea consumption has an inverse association with chronic insomnia-disrupted gut microbiota and bile acids. Our results suggest that the gut microbiota-bile acid axis may be an important preventive target for mitigating the detrimental impact of chronic insomnia on cardiometabolic health.

## Methods

**Description of study design and populations.** We used two human cohorts in the present study: GNHS, as a discovery cohort[37] and GGMP, as a validation cohort[38]. We integrated multi-omics data from the GNHS to investigate whether the gut microbiota-bile acid axis contributed to the positive association between chronic insomnia and CMD, and to explore dietary approaches that could alleviate the association between chronic insomnia and CMD. We then validated these associations in an independent large cross-sectional cohort study: GGMP. The study protocol for the GNHS was approved by the Ethics Committee of the School of Public Health at Sun Yat-sen University and Ethics Committee of Westlake University, and all participants provided written informed consent. The study protocol for the GGMP was approved by the Ethical Review Committee of Chinese Center for Disease Control and Prevention (Beijing, China), and all the participants provided written informed consent.

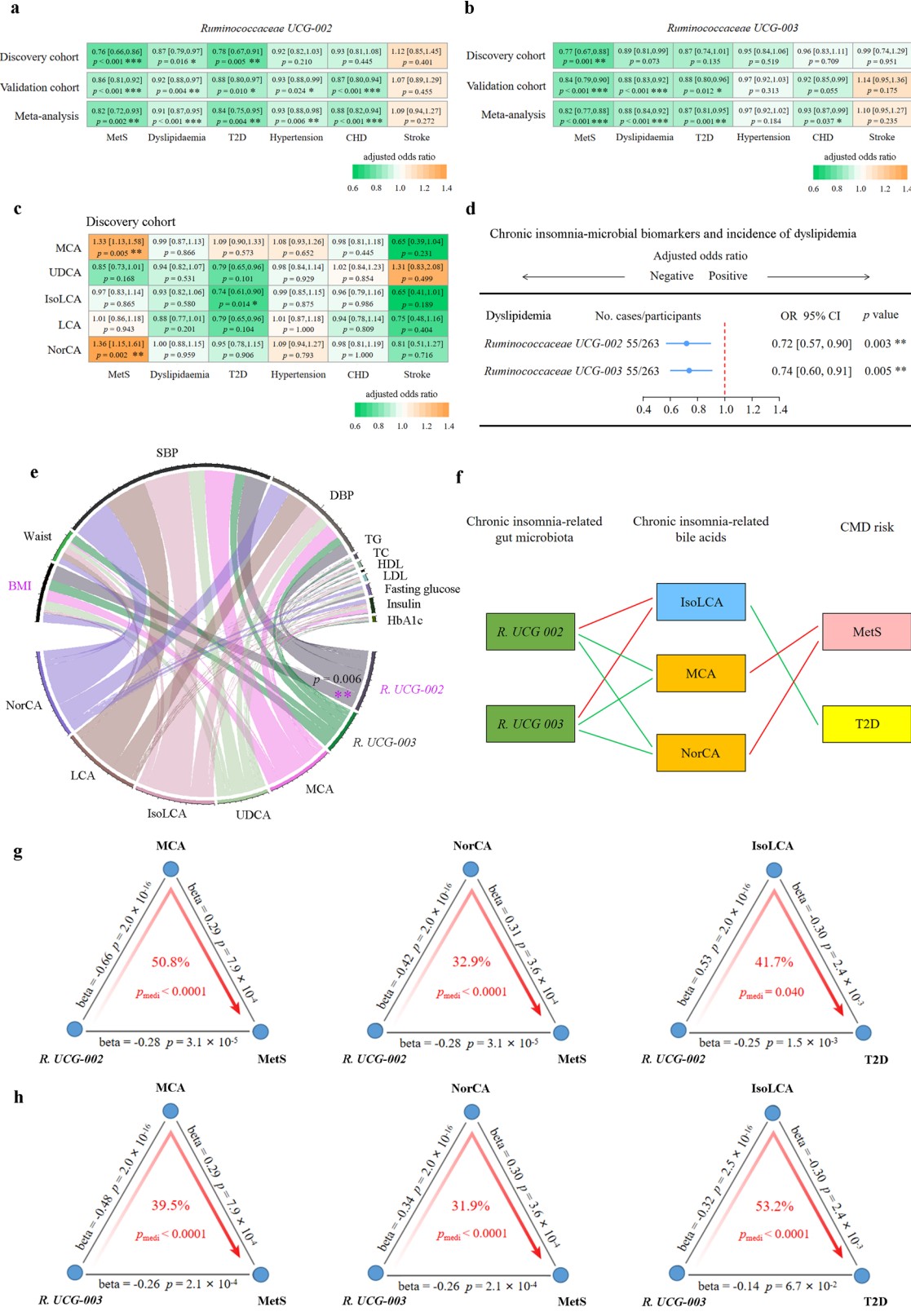

The GNHS was a community-based prospective cohort including 4048 participants of Han Chinese ethnicity[37]. Briefly, a total of 4048 participants, 40–75 years old and living in southern China, Guangzhou City, were recruited into the GNHS between 2008 and 2013. Fecal samples of the participants were collected at the second follow-up at the study site up to Apr 30, 2019 (median follow-up of 6.2 years from entry into the cohort). We excluded participants who were (1) without measurement of gut microbiota data (n = 2125); (2) without valid questionnaire information on chronic insomnia (n = 4); (3) with self-reported baseline cancers, chronic renal dysfunction, or cirrhosis (n = 71); (4) with missing covariates (age, sex, BMI, education, income, smoking status, alcohol status, total energy intake, and physical activity) (n = 24); and (5) with extreme levels of dietary total energy intake (men: <800 kcal or >4000 kcal; women: <500 kcal or >3500 kcal) (n = 15). Finally, 1809 participants were included in the present analysis.

Chronic insomnia was defined as meeting one of the five following criteria for at least three days a week for at least 6 months: (i) taking >30 min to fall asleep, (ii) experiencing nocturnal awakening ≥2 times or early morning awakening, (iii)

**Fig. 3 Association of the chronic insomnia-related gut microbiota-bile acid axis with cardiometabolic diseases.** Multivariable logistic regression was used to estimate the association of the chronic insomnia inverse-related microbial biomarkers *Ruminococcaceae UCG-002* (**a**) and *Ruminococcaceae UCG-003* (**b**) with different CMD in the discovery and validation cohorts, respectively. The effect estimates from the discovery and validation cohorts were pooled using random effects meta-analysis. **c** Multivariable logistic regression was used to estimate the association of the chronic insomnia-related bile acid biomarkers with CMD in the discovery cohort. **d** The prospective associations of the above identified gut microbiota biomarkers (measurement of gut microbiota data at the second follow-up) with the incidence of CMD (dyslipidemia) at the third follow-up using multivariable logistic regression, adjusted for potential confounders. Error bars in **a**–**d** are odds ratios with 95% confidence intervals. **e** Associations of the identified gut microbiota and bile acid biomarkers with CMD-related risk factors (BMI, DBP, SBP, waist circumference, fasting serum TG, TC, HDL, LDL, glucose, insulin, and HbA1c) using multivariable linear regression model in the GNHS, adjusted for potential confounders. **f** Parallel coordinate chart showing the association among gut microbes, bile acid biomarkers and CMD outcomes. The left panel shows the microbial biomarkers, the middle panel shows the bile acid biomarkers, and the right panel shows the CMD outcomes. The red lines across panels indicate the positive association. The green lines across panels indicate the inverse association. **g** The chronic insomnia inverse-related microbial biomarker *Ruminococcaceae UCG-002* affects risk of MetS and T2D though specific bile acid biomarkers, respectively. **h** The chronic insomnia inverse-related microbial biomarker *Ruminococcaceae UCG-003* affects the risk of MetS and T2D through specific bile acid biomarkers, respectively. The gray lines indicate the associations, with corresponding normalized beta values and *p* values. The red arrowed lines indicate the microbial effects on CMD mediated by specific bile acid biomarkers, with the corresponding mediation *p* values. *p* value < 0.05 is considered significantly different. Throughout the above analyses, FDR from multiple testing was controlled by the Benjamini-Hochberg method. All statistical tests were two-sided. Source data are provided as a Source Data file. CMD cardiometabolic disease; T2D type 2 diabetes; MetS metabolic syndrome; SBP systolic blood pressure; DBP diastolic blood pressure; TG triglycerides; TC total cholesterol; HDL high-density lipoprotein cholesterol; LDL low-density lipoprotein cholesterol; HbA1c glycated hemoglobin.

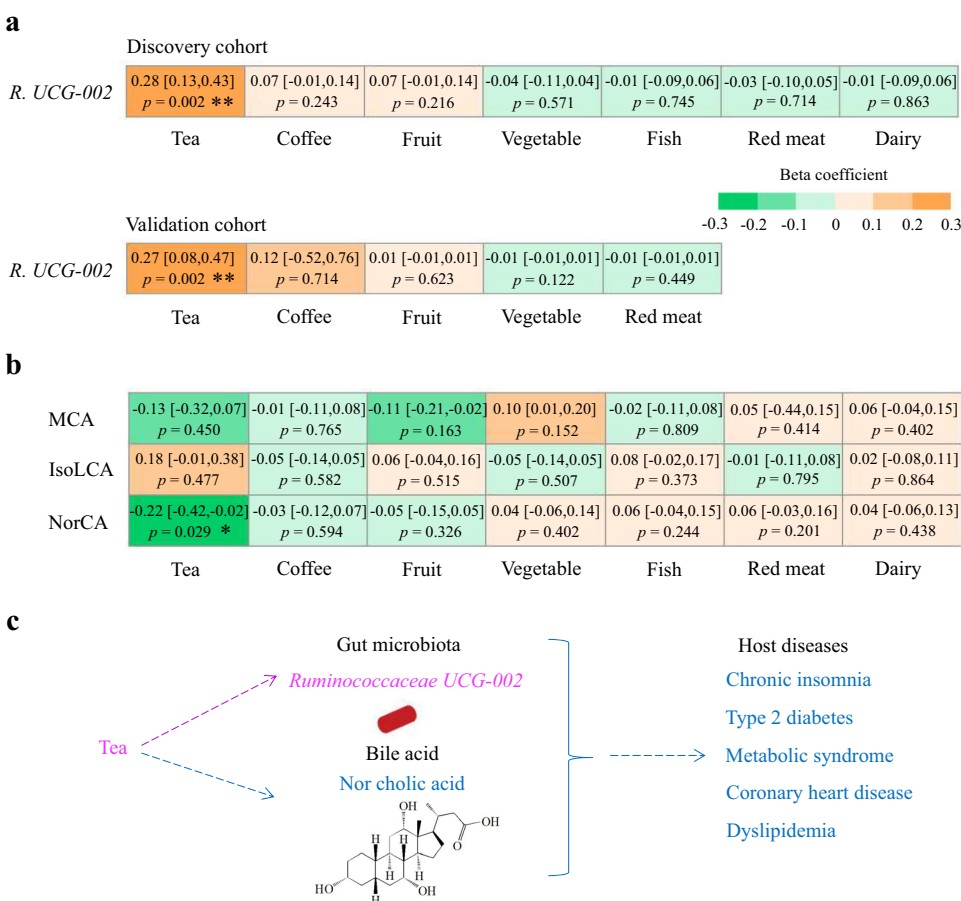

**Fig. 4 Habitual dietary intake and gut microbiota-bile acid axis. a** Prospective association of dietary factors with the identified microbial biomarker *Ruminococcaceae UCG-002* in the discovery and validation cohorts. The results of *Ruminococcaceae UCG-003* are reported in the Supplementary Fig. 7. Values presented are beta coefficients (95% confidence intervals) with corresponding *p*-values. **b** Prospective association of dietary factors with the identified bile acid biomarkers linking chronic insomnia and cardiometabolic diseases (CMD) in the GNHS. Multivariable linear regression was used to examine the prospective association of dietary factors with microbial and bile acid biomarkers, adjusted for the potential confounders. *p* value < 0.05 is considered significantly different. Value presented are beta coefficients (95% confidence intervals) with corresponding *p*-values. **c** Diagram of the link between habitual tea consumption, the gut microbiota-bile acid axis, and CMD. Habitual tea consumption was associated with higher abundance of *Ruminococcaceae UCG-002* and lower abundance of the Nor cholic acid (NorCA). Value with asterisk is significantly different. (*p* < 0.05, **p < 0.01, ***p < 0.001). Throughout the above analyses, FDR was controlled by the Benjamini-Hochberg method. All statistical tests were two-sided. Source data are provided as a Source Data file.

**Table 3 The stratified analysis of the association of *Ruminococcaceae UCG-002* with CMD risk by tea consumption (yes versus no) in the Guangzhou Nutrition and Health Study[a].**

| CMD | Odds ratio (OR) | 95% CI | *p* value |
|---|---|---|---|
| Tea consumption group (*n* = 928) | | | |
| dyslipidemia | 0.85 | [0.74, 0.98] | 0.024 |
| T2D | 0.73 | [0.60, 0.89] | 0.002 |
| MetS | 0.79 | [0.67, 0.93] | 0.006 |
| Non-tea consumption group (*n* = 791) | | | |
| dyslipidemia | 0.90 | [0.78, 1.04] | 0.168 |
| T2D | 0.84 | [0.65, 1.07] | 0.162 |
| MetS | 0.70 | [0.55, 0.86] | 0.001 |

*T2D* type 2 diabetes, *MetS* metabolic syndrome.
[a]Multivariable logistic regression (odds ratio) was used to estimate the association of tea consumption with cardiometabolic disease (CMD) risk, adjusted for the potential covariates. The Benjamini-Hochberg method was used to control the false discovery rate (FDR) for multiple testing. All statistical tests were two-sided.

having light sleep and dreaminess, (iv) experiencing total sleep time <6 h and (v) having daytime symptoms such as fatigue, attention deficits or mood instability, according to the Chinese Clinical Expert Consensus on the definition, diagnosis and medical treatment of insomnia[73].

The GGMP participants were divided into the following two groups according to the criteria for chronic insomnia for at least 3 days a week for at least 1 month: (i) Non-chronic insomnia group, and (ii) Chronic insomnia group.

For both cohorts, T2D was defined as fasting blood glucose ≥ 7.0 mmol/L, HbA1c ≥ 6.5% or diabetic medication[74]. Hypertension was defined as systolic blood pressure (SBP)/diastolic blood pressure (DBP) ≥ 140/90 mmHg or medical history[75]. Dyslipidemia was defined as total cholesterol (TC) ≥ 6.2 mmol/L or triglycerides (TG) ≥ 2.3 mmol/L or low density lipoprotein cholesterol (LDL) ≥ 4.1 mmol/L or high density lipoprotein cholesterol (HDL) < 1.0 mmol/L or medical history[76]. MetS was defined as meeting three of the five following criteria: (i) waist > 90 cm (male) or waist >85 cm (female), (ii) fasting blood glucose (FBG) ≥ 6.1 mmol/L (110 mg/dl) or previously diagnosed with T2D, (iii) TG ≥ 1.7 mmol/L (150 mg/dl), (iv) HDL < 1.04 mmol/L (40 mg/dl) and (v) SBP/DBP ≥ 130/85 mmHg or previously diagnosed with hypertension[76]. CHD and stroke were determined by self-report (conformed with previous diagnosis).

**Metadata collection in the GNHS.** For the GNHS, during the on-site face-to-face questionnaire interviews, we collected information on sociodemographic, lifestyle, dietary factors, and medical history. Anthropometric parameters, including weight, height, waist circumference, and hip circumference, were measured by trained staff. Total energy intake was calculated according to the Chinese Food Consumption Table, 2002[77]. Physical activity was assessed as the total metabolic equivalent for task hours per day based on a questionnaire for physical activity[78].

Fasting venous blood samples were taken at the recruitment and follow-up visits and were aliquoted and stored in a −80 °C freezer prior to analysis. FBG, TG, TC, HDL, and LDL were measured by colorimetric methods using a Roche Cobas 8000 c702 automated analyzer (Roche Diagnostics GmbH, Shanghai, China). Insulin was measured by electrochemiluminescence immunoassay methods using a Roche Cobas 8000 e602 automated analyzer (Roche Diagnostics GmbH, Shanghai, China). High-performance liquid chromatography was used to measure the HbA1c level using the Bole D-10 Hemoglobin A1c Program on a Bole D-10 Hemoglobin Testing System.

**Fecal sample collection, DNA extraction, and 16S rRNA gene sequencing in the GNHS.** During a follow-up visit to the study center, the participants were given a stool sampler and provided detailed instructions for stool sample collection. Briefly, each participant collected their stool sample after defecation and gave the sample to the staff immediately. The stool samples with ice bags were transported to the research laboratory and stored in a −80 °C freezer within 4 h. Detailed information regarding DNA extraction and gut microbiota 16S rRNA gene sequencing in the GNHS is provided in the Supplementary methods.

**Targeted fecal bile acid profiling in the GNHS.** Targeted bile acid profiling of fecal samples (*n* = 954) was performed with an ultra-performance liquid chromatography coupled to tandem mass spectrometry (UPLC-MS/MS) system (ACQUITY UPLC-Xevo TQ-S, Waters Corp., Milford, MA, USA) at Metabo-Profile Biotechnology Co., Ltd. (Shanghai, China) (Supplementary methods).

**Description of the GGMP.** The GGMP is a large community-based cross-sectional cohort conducted between 2015 and 2016 including 7009 participants with high quality gut microbiome data[38]. The GGMP participants were from 14 randomly selected districts or counties in Guangdong Province, China. In face-to-face questionnaire interviews, the host metadata, including socio-demographic features, disease status, lifestyle and dietary information, were collected. We excluded participants (1) without chronic insomnia information (*n* = 633); and (2) with missing covariates (age, sex, BMI, education, smoking status, alcohol status) (*n* = 254). Finally, we included 6122 participants (52.8 ± 14.7 y, 55.2% of women) from the GGMP in our analysis as an independent validation cohort. The characteristics of the included participants in the GGMP are presented in Table 2. Detailed information regarding the host metadata and stool sample collection and the 16S rRNA gene sequencing process in the GGMP have been reported previously[38].

**Statistical analysis.** We compared differences between four groups using the chi-square test for categorical variables and ANOVA for continuous variables. In the GNHS, we examined the association of chronic insomnia with gut microbial α-diversity indices (Observed species, Shannon index, Chao 1 index, ACE index and Simpson index) among the four groups using a multivariable linear regression with three different statistical models. Model 1 was adjusted for age, sex, BMI, smoking status, alcohol status, physical activity, education, income and total energy intake at baseline. Model 2 was additionally controlled for hypertension, hyperlipidemia, MetS, T2D, CHD, stroke, and medication for T2D. Model 3 was further adjusted for dietary intake of vegetables, fruits, red and processed meat, fish, dairy products, coffee and tea. The association between chronic insomnia and β-diversity dissimilarity based on genus-level Bray-Curtis distance was examined using permutational ANOVA (PERMANOVA) (999 permutations).

In the GNHS, there was a significant difference in the gut microbial structure for the New-onset group or Long-term chronic insomnia group, compared with the Long-term healthy group (Fig. 1b and Supplementary Fig. 1). To identify robust microbial biomarkers of chronic insomnia and increase the sample size, we combined the New-onset group and Long-term chronic insomnia group into Chronic insomnia group. We used MaAsLin to identify potential chronic insomnia associated gut microbiota (*q* value < 0.25 was used as the threshold of significance in the exploratory analyses, as commonly used previously[26,79]) using the above three different statistical models by comparing the Chronic insomnia group with the Long-term healthy group. The Benjamini-Hochberg method was used to control the false discovery rate (FDR).

Next, we used OPLS-DA to identify potential bile acids associated with chronic insomnia. We further used linear regression, adjusted for the same covariates as above model 3, to confirm the association of chronic insomnia with the OPLS-DA selected bile acids. Given that dietary cholesterol intake and fiber intake might be potential confounders affecting the relationship between chronic insomnia and the bile acid pool, we further performed a sensitivity analysis by including dietary cholesterol intake and fiber intake as additional covariates in the above model 3. We examined the association of the above identified gut microbiota biomarkers with bile acid biomarkers using partial correlation analysis, adjusted for age, sex and BMI. In addition, we tested the association of chronic insomnia with another two important classes of gut microbial metabolites (short-chain fatty acids, aromatic amino acids and their derivatives) by using multivariable linear regression, adjusted for the same covariates as above in model 3. The Benjamini-Hochberg method was used to control FDR.

To gain further mechanistic insight into the connection between the chronic insomnia and CMD, we investigated the correlation of the chronic insomnia-related microbial and bile acid biomarkers with different CMD and risk factors (BMI, DBP, SBP, waist circumference, and fasting serum levels of TG, TC, HDL, LDL, glucose, insulin, and HbA1c) using multivariable logistic regression and linear regression model in the GNHS, adjusted for age, sex, smoking status, alcohol status, physical activity, education, income, and total energy intake. We further examined the prospective association of the above identified gut microbiota biomarkers with the incidence of CMD outcomes at the third follow-up using multivariable logistic regression, adjusting for age, sex, smoking status, alcohol status, physical activity, education, income, and total energy intake. Throughout the above analyses, correction of multiple testing was conducted by using the Benjamini-Hochberg method.

Based on the biological plausibility of the associations among the gut microbiota, bile acids and CMD[54,59,80,81], and our above findings, we performed mediation analysis to evaluate whether bile acids could mediate the association of the chronic insomnia related-gut microbiota with CMD outcomes (gut microbiota → bile acids → CMD). The mediation analysis was performed to examine the mediating effect of bile acids on the association of the chronic insomnia-related gut microbiota with CMD outcomes[82,83]. We defined three pathways in the mediation analysis: (1) exposure to mediator; (2) mediator to outcome; and (3) exposure to outcome. In the mediation analysis, the covariates included age, sex, BMI, smoking status, alcohol status, physical activity, education, income, and total energy intake. The mediation analysis was performed using the R package "mediation" with the same parameter settings (boot = "TRUE", boot.ci.type = "perc", conf.level = 0.95, sims = 1000). The total effect was obtained through the sum of a direct effect and a mediated (indirect) effect. The percentage of the mediated effect was calculated using the formula: (mediated effect/total effect) × 100. Sensitivity analysis was

performed to test the robustness of the mediation effect and violation of the assumption (sequential ignorability) using the R package "medsens" with default parameters[84,85]. The reporting of the mediation results followed the Guideline for Reporting Mediation Analyses (AGReMA) statement[86].

Finally, we used a linear regression model to determine the prospective association of dietary factors with the gut microbial and bile acid mediators of chronic insomnia and CMD, adjusted for age, sex, BMI, smoking status, alcohol status, physical activity, education, income, dietary intake of vegetables/fruits/red and processed meat/fish/dairy products/coffee/tea) (mutual adjustment for each other) and total energy intake. The analyses were conducted among the GNHS participants without chronic insomnia or CMD at baseline.

In the GGMP participants, we used a multivariable linear regression model to examine the association of chronic insomnia with the gut microbiota structure and the identified gut microbiota biomarkers, adjusting for age, sex, BMI, smoking status, alcohol status, education, dietary intake of vegetables, fruits, and red and processed meat. We conducted a secondary analysis to evaluate the association of the insomnia symptoms score (per unit change) with the identified gut microbiota biomarkers by using linear regression, adjusted for the same covariates. We also used logistic regression and linear regression to examine the association between the gut microbiota biomarkers and different CMD outcomes, adjusted for age, sex, smoking status, alcohol status, and education. For the GGMP participants, we did not include income in the statistical models due to a large number of missing values (income data were available among 3774 out of 6122 participants). We therefore performed a sensitivity analysis with further adjustment for income in the above analyses to examine the robustness of the models. Then, for each of the above linear regressions or logistic regressions, the effect estimates from the GNHS and the GGMP were pooled by random effects meta-analysis. In addition, we further performed additional interaction analysis and stratified analyses by age and sex to explore potential heterogeneity for the chronic insomnia-gut microbiota association and the gut microbiota-CMD association, and used random effects meta-analysis to pool the effect estimates from the GNHS and GGMP.

In the GGMP, we also used multivariable linear regression to examine the association of dietary factors with gut microbial features of chronic insomnia and CMD, adjusted for age, sex, BMI, smoking status, alcohol status, education, dietary intake of vegetables/fruits/red and processed meat/tea/coffee (mutual adjustment for each other). The analyses were conducted among the GGMP participants without chronic insomnia or CMD. We also performed additional stratified analyses by tea consumption (yes versus no) using logistic regression in the GNHS to explore whether the associations between the chronic insomnia-related gut microbiota and CMD risk factors could be affected by tea consumption. We further investigated the association of tea consumption with the risk of chronic insomnia using logistic regression in the GNHS and GGMP and used random effects meta-analysis to pool the effect estimates from the GNHS and GGMP.

In the GNHS, we used the co-occurrence network analysis based on the above partial correlation coefficient to demonstrate the interaction of the above gut microbiota and bile acid biomarkers, and only the significant correlations (larger than 0.1 or smaller than −0.1) were used for network construction. The networks were further visualized in Cytoscape software version 3.7.2. We used R version 3.6.3 for statistical analysis unless otherwise specified, and $p$ value < 0.05 was considered statistically significant.

**Reporting summary**. Further information on research design is available in the Nature Research Reporting Summary linked to this article.

## Data availability

16S rRNA gene sequencing data of the Guangzhou Nutrition and Health Study (GNHS) are available in the Genome Sequence Archive (GSA) (https://ngdc.cncb.ac.cn/gsa/) at accession number CRA006769. 16S rRNA gene sequencing data of the Guangdong Gut Microbiome Project (GGMP) are available from the European Nucleotide Archive (https://www.ebi.ac.uk/ena/) at accession number PRJEB18535. The Sliva reference database version 138 was used to annotate taxonomic information. The metadata of the GGMP are available in a previous publication (https://pubmed.ncbi.nlm.nih.gov/30250144/)[38]. The data associated with this study are presented in the paper, supplementary information and Source Data file. Source data are provided with this paper.

## Code availability

Codes used for this study are available at: https://github.com/nutrition-westlake/Chronic-insomnia-Gut-microbiota-bile-acid-axis-and-Cardiometabolic-diseases-Project/blob/main/Code%20available.

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

## Acknowledgements

We thank the Westlake University Supercomputer Center for computational resources and related assistance and all the participants involved in the Guangzhou Nutrition and Health Study and the Guangdong Gut Microbiome Project. This study was funded by the National Natural Science Foundation of China (82073529, 81903316, 81773416), Zhejiang Ten-thousand Talents Program (2019R52039), Zhejiang Provincial Natural Science Foundation of China (LQ19C200005, LQ21H260002), Westlake Education Foundation and the 5010 Program for Clinical Research (2007032) of Sun Yat-sen University (Guangzhou, China).

## Author contributions

J.S.Z. and Y.M.C. designed the study and developed the concept; F.Z.X., W.L.G., Z.L.M., and M.L.S. collected the data; Y.H.L., C.M.X., X.X.L., Y.Y.T., and J.L.W. processed the samples; Z.L.J., L.B.Z., Y.H., Y.Q.F., L.Q.S., J.T., and K.D. analyzed the data; Z.L.J., L.B.Z., Y.H., J.S.Z., Y.M.C., and H.W.Z. drafted the manuscript; J.S.Z., Y.M.C., and H.W.Z. obtained the funding; and all authors reviewed and revised the final manuscript.

## Competing interests

All authors declare that they have no competing interests.
