## [Peer Review File · Nature Communications]

Reviewer comments, first round

Reviewer #1 (Remarks to the Author):

Summary

• In this manuscript, Jiang, Zhuo, and He et al. observed in two large, independent human studies that subjects suffering from insomnia had different gut microbiomes than healthy controls. Furthermore, the bile acid pool and risk for CMD were altered in insomniacs. While the size of the human sample pool is impressive, the conclusions in this manuscript are repeatedly overstated and, in several instances, are not supported by the data. This manuscript lacks any causative data which dampens overall enthusiasm for publication. Specific recommendations are outlined below.

Major Issues

- There is a general oversimplification linking alterations in gut microbiome composition to alterations in bile acids (i.e. microbiome = bile acids). However, alterations in gut microbial communities will alter many metabolites beyond bile acids. Secondary bile acids are one of likely thousands of small molecule metabolites produced by commensal bacteria, and a more balanced discussion and investigation is needed of other microbe-associated metabolic pathways.
- Although the authors acknowledge this as a limitation in the discussion, the data presented in this manuscript are purely associative and descriptive. No causative evidence is presented—this severely dampens enthusiasm for publication.
- Associations are incorrectly reported as causation in several instances or conclusions are drawn based purely on correlative analysis.
 - o Lines 206-208, 226-230 and 259-262.
- The differences reported in the bile acid pool are not striking. Although the authors corrected for several factors including some dietary substrates, more specific corrections for dietary cholesterol intake or dietary fiber intake could possibly mitigate any significance in the bile acid pool between healthy human subjects and those with insomnia.
- The authors conclude, "Our results from two large cohort studies provided timely evidence supporting the effect of chronic insomnia on the gut microbiome". This conclusion is not supported by the data as it implies directionality. No evidence was presented to suggest that insomnia causes a change in the gut microbiome when the inverse could also be true.
- The connection to insomnia is diluted progressively throughout the manuscript. By Figure 4, the data presented are only tangentially related to insomnia. The manuscript does not flow in a logical progression.

Minor Issues

- Abstract: The authors state "Chronic insomnia is the second most prevalent mental disorder..." It is understood that insomnia is associated with other mental disorders in some cases, but it seems incorrect to list insomnia itself as a mental disorder.
- For all alpha diversity figures, the text mentions that the figure is Shannon and Chao1 indices, but it remains unclear which one is actually shown. What about ACE and Simpson indices?
- Figure 2 a-c and e, exact p/q values should be shown instead of asterisk. This should be applied to entire manuscript.
- The authors report that Ruminococcaceae UCG-002 and Ruminococcaceae UCG-003 are significantly negatively associated with insomnia. The authors perform subsequent correlations and report various conclusions under this assumption. However, these data are based on the very liberal cutoff of $* q < 0.25$ in Fig. 2c and Supplementary Table 3. The cutoff of $* q < 0.25$ in Fig. 2c and Supplementary Table 3 is not appropriate. This indicates that one quarter of values indicated as significant will be false positive. Maximum of $q < 0.05$ should be used.
- Figure 2e—Insomnia group is shown before control, this should be reversed.
- Figure 3a and b—the association with stroke is not significant? This is why exact p values need to be shown.
- Figure 3f—this is not an appropriate use of a Sankey diagram. The purpose of a Sankey is to show proportionality of flow.
- The conclusion that tea consumption was significantly associated with increased levels of R. UCG-002 is interesting but the authors failed to state whether or not tea consumption was

associated with lower levels of insomnia.

- This manuscript would require significant English language revisions.

Reviewer #2 (Remarks to the Author):

This research studies whether gut microbiota plays a role in the association between chronic insomnia and cardiometabolic diseases (CMD). Two microbial mediators are identified in a longitudinal cohort and replicated in another independent cohort. Mediator role of bile acids for the association between gut microbiota and CMD is examined using bi-directional mediation analysis. Microbiota-bile acid axis may be a potential intervention target to diminish the impact of chronic insomnia on cardiometabolic health.

This manuscript is well organized and well written. The study groups involved are large while one group is used as replication. I have some minor comments as follows.

Line 102: why "In the GGMP, participants were divided into two groups: (i) Non-chronic insomnia group, and (ii) Chronic insomnia group (Methods)" while in the GNHS participants were divided into four?

Line 128: Is "Multivariate Analysis by Linear Models (MaAsLin)" the same as a linear regression with multiple predictors? Is chronic insomnia the outcome variable? If so, shouldn't logistic regression be used given chronic insomnia is dichotomous?

Lines 139-150: What are the results obtained from the "orthogonal partial least squares discrimination analysis (OPLS-DA)"? From which method (OPLS-DA or logistic regression) these p-values are?

Line 152: In this section it would be nice to mention the statistical method used, like in the previous sections.

Line 215: change "relate" to "related"?

Reviewer #3 (Remarks to the Author):

This is an interesting and important study to examine the associations of insomnia with gut microbiome and its role in cardiometabolic disease risk. Overall, the study is well designed and sufficiently powered, and the analysis appears appropriate. Below are my comments/suggestions to further improve the manuscript.

- Introduction: the authors need to provide stronger rationales for their study in the introduction. For example, the concept of the brain-gut axis should be mentioned. Also, 1-2 sentences could be added to summarize prior studies showing that gut microbiomes also exhibit circadian rhythms, which interact with the host circadian rhythms, and sleep disturbances such as chronic insomnia could in turn disrupt microbial circadian rhythms, thus influencing gut microbial composition and activity.

- It seems that the definition and assessment of chronic insomnia were somewhat different between the discovery and validation cohort. What impact do the authors think this difference may have on the results?

- Another common way to assess insomnia is to use a symptom score to characterize the severity of the disease. The authors could conduct a secondary analysis evaluating the number of insomnia symptoms in relation to the gut microbial taxa/metabolites identified in the primary analysis. If a dose-response relationship is found, this could provide further support to their study conclusion.

- The authors claimed that "tea consumption may alleviate the detrimental impact of chronic insomnia on the cardiometabolic health". However, this suggestion was not directly supported by the results presented. The authors can perform a stratified analysis by tea consumption to examine whether the associations between insomnia-related gut microbiomes and CMD risk factors are weaker among participants with high habitual tea consumption versus those with low

consumption.

- For the primary findings, additional stratified analyses (at least by age and sex) are needed to explore potential heterogeneity in the association (it is known that both insomnia and CMD risk are strongly related to age and sex).

- Figure 1 can be improved by clearly indicating when stool samples were collected, when insomnia assessment was done and repeated (two time points), and when cardiometabolic risk factors were assessed. Such information can help readers better understand the study design.

- Supplemental Table 2: I would suggest presenting the sample characteristics by chronic insomnia status in the validation cohort too, similar to the discovery cohort. It would be helpful to compare the prevalence of chronic insomnia and understand any potential differences between the two study cohorts.

- Discussion: The findings on secondary bile acids are very interesting. Some prior studies also support these findings, but the authors did not include them in their discussion. For example, in one study that examined insomnia symptoms and plasma metabolomics (PMID: 30371783), derivatives of second bile acids (e.g., glycoursodeoxycholic acid) were positively associated with more severe insomnia symptoms. Another study found that Ruminococcaceae was related to plasma LDL and triglycerides, which has direct implications for cardiovascular risk (PMID: 31862950).

- Discussion: Sleep-disordered breathing often leads to chronic insomnia and poor sleep quality. Intermittent hypoxia from sleep-disordered breathing may also have an impact on gut microbiome, as suggested by some recent evidence (PMID: 33705556, 29896566, 26711739). As sleep-disordered breathing was not considered in the current study, the authors should discuss this and acknowledge this as a limitation (or a potential explanation to their findings)

Discussion: Given that caffeine in tea may exacerbate insomnia, the authors need to be cautious when interpreting tea consumption as a potential intervention strategy to reduce the adverse cardiometabolic impact of insomnia-related gut microbial alterations.

Reviewer #4 (Remarks to the Author):

The authors investigated the associations between chronic insomnia, bile acid, and CMD. The authors performed a mediation analysis to uncover plausible mechanisms of microbial effects on CMD through specific bile acids. I commend the authors on applying mediation analysis to investigate mechanisms of chronic insomnia on CMD, however their analyses fall short on a few important points mentioned below. Overall, the authors need a more careful consideration of the causal assumptions for mediation analysis and clearer reporting of the methods used.

Major Comments:

It is unclear what the authors mean with the term "bi-directional" mediation analysis. This is not a commonly used term for mediation analysis. Similarly, the term "inverse" mediation is not commonly used.

Most importantly, it appears the authors tried to demonstrate evidence for mediation of microbial effects -> bile acid -> CMD as opposed to mediation microbial effects -> CMD -> bile acid. There are two major limitations to this. First, there are six different plausible models of the three variables – the authors only investigated two plausible models. Second, and the most important, comparing the statistical significance and/or magnitude of a mediated effect estimate from one model (e.g., microbial effects -> bile acid -> CMD) to the statistical significance and/or magnitude of a mediated effect estimate from another model (e.g., microbial effects -> CMD -> bile acid) does not provide evidence for mediation through one mechanism over the other. These models are from the same equivalence class and therefore cannot be distinguished from one another by statistical tests alone. There must be a compelling scientific reason to suspect that these are the two plausible mediation models and even if there is compelling scientific rationale for these two models, statistics cannot distinguish between these two models. Please see Thoemmes (2015) for more details.

Thoemmes, F. (2015). Reversing arrows in mediation models does not distinguish plausible models. *Basic and Applied Social Psychology*, 37(4), 226–234.
<https://doi.org/10.1080/01973533.2015.1049351>

It is unclear how the authors estimated the mediated effects. There are many statistical methods used in the mediation literature to estimate mediated effects and perform significance testing of mediated effects and it is important that the authors clarify this. Additionally, the authors did not mention the necessary causal assumptions that are required when performing mediation analysis. The interpretation of mediated effects relies on specific no unmeasured confounding assumptions.

MacKinnon, D. P. (2008). *Introduction to statistical mediation analysis*. Routledge.

Vanderweele, T. J. (2015). *Explanation in Causal Inference: Methods for Mediation and Interaction* (Vol. 53). <https://doi.org/10.1017/CBO9781107415324.004>

Valente, M. J., Rijnhart, J. J. M., Smyth, H. L., Muniz, F. B., & Mackinnon, D. P. (2020). Causal Mediation Programs in R , Mplus , SAS , SPSS , and Stata. *Structural Equation Modeling: A Multidisciplinary Journal*, 00(00), 1–10. <https://doi.org/10.1080/10705511.2020.1777133>

There are also sensitivity analyses that are designed to assess the robustness of mediated effects to violations of these assumptions. For example of a sensitivity analysis method see:

Cox, M. G., Kisbu-Sakarya, Y., Miocevic, M., & MacKinnon, D. P. (2013). Sensitivity Plots for Confounder Bias in the Single Mediator Model. *Evaluation Review*, 37(5), 405–431. <https://doi.org/10.1177/0193841X14524576>

Imai, K., Keele, L., & Yamamoto, T. (2010). Identification, Inference and Sensitivity Analysis for Causal Mediation Effects. *Statistical Science*, 25(1), 51–71. <https://doi.org/10.1214/10-STS321>

Smith, L. H., & Vanderweele, T. J. (2019). Mediation E-values: Approximate sensitivity analysis for unmeasured mediator-outcome confounding. *Epidemiology*, 30(6), 835–837. <https://doi.org/10.1097/EDE.0000000000001064>

Vanderweele, T. J. (2010). Bias formulas for sensitivity analysis for direct and indirect effects. *Epidemiology*, 21(4), 540–551. <https://doi.org/10.1097/EDE.0b013e3181df191c>

The authors would be encouraged to follow the AGRReMA statement for the reporting of mediation results. The following reference contains the necessary reporting guidelines for mediation analysis and many key citations that I would highly encourage the authors consult.

Lee, H., Cashin, A. G., Lamb, S. E., Hopewell, S., Vansteelandt, S., VanderWeele, T. J., ... & Henschke, N. (2021). A Guideline for Reporting Mediation Analyses of Randomized Trials and Observational Studies: The AGRReMA Statement. *JAMA*, 326(11), 1045–1056.

Dear reviewers,

RE NCOMMS-21-28734-T

Chronic insomnia, gut microbiota-bile acid axis and cardiometabolic health: results from two large-scale human cohorts

We would like to thank you for your helpful comments and suggestions for improving our manuscript. We have uploaded a clean version of the revised manuscript and a version with changes highlighted in yellow colour. Our point-by-point responses to your comments are presented below.

Reviewer #1:

Summary: In this manuscript, Jiang, Zhuo, and He et al. observed in two large, independent human studies that subjects suffering from insomnia had different gut microbiomes than healthy controls. Furthermore, the bile acid pool and risk for CMD were altered in insomniacs. While the size of the human sample pool is impressive, the conclusions in this manuscript are repeatedly overstated and, in several instances, are not supported by the data. This manuscript lacks any causative data which dampens overall enthusiasm for publication. Specific recommendations are outlined below.

Response: We thank you for your helpful comments. We took your suggestions and modified our conclusions to tone down our claims to avoid potential overstatement. We also revised the related statements and discussion about the causality throughout the text. Please see our responses to your specific comments below:

1. There is a general oversimplification linking alterations in gut microbiome composition to alterations in bile acids (i.e. microbiome = bile acids). However, alterations in gut microbial communities will alter many metabolites beyond bile

acids. Secondary bile acids are one of likely thousands of small molecule metabolites produced by commensal bacteria, and a more balanced discussion and investigation is needed of other microbe-associated metabolic pathways.

Response: We agree with you that there are much more metabolites in addition to bile acids in the gut. In our prior version of the manuscript, we mainly focused on the analysis of bile acids as a pre-defined strategy. This is because bile acids are very important biologically active metabolites closely related with both gut microbiome and human metabolic health. Therefore, we hypothesized that the gut microbiota-bile acid axis may potentially play a role linking chronic insomnia and cardiometabolic diseases.

Nevertheless, as you pointed out, alterations in gut microbiota would alter many other metabolites. Thus, as a secondary analysis in the revision of the manuscript, we did additional analyses to estimate the association of chronic insomnia with another two important classes of gut microbe-associated metabolites: short chain fatty acids, aromatic amino acids and their derivatives. We found that chronic insomnia was not associated with short chain fatty acids, aromatic amino acids or their derivatives in our cohort.

We have now added the additional results in the Supplementary Table 7 and in the text of the revised manuscript. We also added a more balanced discussion about this point as suggested in the discussion section (lines 273-278):

Lines 168-170: *“In addition, chronic insomnia was not associated with short-chain fatty acids, aromatic amino acids or their derivatives (Supplementary Table 7).”*

Lines 671-675: *“In addition, we tested the association of chronic insomnia with another two important classes of gut microbial metabolites (short-chain fatty acids, aromatic amino acids and their derivatives) by using multivariable linear regression, adjusted for the same covariates as above model 3.”*

Lines 273-278: *“As alterations in gut microbiota would alter many other metabolites, we did a secondary analysis for the association of chronic insomnia with short-chain fatty acids, aromatic amino acids and their derivatives^{46,47}. Although we did not find any significant association for these two classes of metabolites, we could not rule out the possibility that chronic insomnia may be associated with other metabolites, which needs further investigations.”*

2. Although the authors acknowledge this as a limitation in the discussion, the data presented in this manuscript are purely associative and descriptive. No causative evidence is presented—this severely dampens enthusiasm for publication.

Response: With respect, we understand your concern about the “causality” evidence. However, we want to argue that epidemiological study with prospective study design such as ours is very important and critical for the evidence-based medicine and translation of basic research. For a topic such as our present study to link chronic insomnia, gut microbiota and cardiometabolic diseases in human, it is almost impossible to run a randomized control trial due to the ethical issue and feasibility issue, therefore, we have to rely on the prospective study to assess the temporal relationship between chronic insomnia and cardiometabolic diseases and then assess the role of gut microbiota-bile acid axis. From this kind of prospective study, we can generate high-level evidence to infer causality and propose novel mechanism directly in humans. We also agree that using animal model may provide evidence of causality and related mechanism, but there is usually a huge gap between animal research and human clinical research, especially in the field of gut microbiome. There are big differences in the gut microbiome between mice models (even for humanized gnotobiotic mouse model) and humans.

Taken together, we believe that our observation is novel with important clinical implications and translational value. As you noted, we have added the discussion about the causality in our limitation section.

-
3. Associations are incorrectly reported as causation in several instances or conclusions are drawn based purely on correlative analysis.
 - o Lines 206-208, 226-230 and 259-262.

Response: As suggested, we have now revised the related sentences to tone down our claims and avoid potential confusion caused by the previous causal language:

Lines 243-246: *“The results indicated that habitual tea consumption was associated with the gut microbiota-bile acid axis, which may potentially underlie the association between chronic insomnia and CMD (Fig. 4c).”*

Lines 264-268: *“Our results from two large cohort studies provided timely evidence supporting that chronic insomnia was associated with the variation of gut microbiome. Specifically, Ruminococcaceae UCG-002 and Ruminococcaceae UCG-003 were identified as the two potential genera inversely associated with chronic insomnia, and these microbes may be associated with host glucose homeostasis and lipid metabolism^{40,41}.”*

Lines 311-313: *“We found that habitual tea consumption was prospectively associated with the identified gut microbiota and bile acids in an opposite direction compared with chronic insomnia.”*

4. The differences reported in the bile acid pool are not striking. Although the authors corrected for several factors including some dietary substrates, more specific corrections for dietary cholesterol intake or dietary fiber intake could possibly mitigate any significance in the bile acid pool between healthy human subjects and those with insomnia.

Response: Thank you for the suggestion. In our previous version, we did not use dietary cholesterol intake or fiber intake as covariates in the statistical models. To

address this reviewer's concerns, we have done a sensitivity analysis to further adjust for dietary cholesterol intake and fiber intake based on our main model 3, and the results showed that adding dietary cholesterol intake and fiber intake into the model did not substantially change the results. The Benjamini-Hochberg method was used to control the false discovery rate (FDR) for multiple testing.

In the GNHS, sensitivity analysis about the inclusion of dietary cholesterol intake and fiber intake as additional covariates in our statistical model showed that chronic insomnia was associated with higher levels of merocholic acid (MCA, β : 0.21, 95%CI: [0.04, 0.38], $p=0.049$) and norcholic acid (NorCA, β : 0.21, 95%CI: [0.04, 0.37], $p=0.042$), and associated with lower levels of isolithocholic acid (IsoLCA, β : -0.26, 95%CI: [-0.43, -0.09], $p=0.030$), lithocholic acid (LCA, β : -0.21, 95%CI: [-0.38, -0.04], $p=0.035$) and ursodeoxycholic acid (UDCA, β : -0.22, 95%CI: [-0.39, -0.05], $p=0.049$).

We have added these new results in the Supplementary Table 6 and in the revised manuscript:

Lines 161-164: *“The results of sensitivity analysis showed that adding dietary cholesterol intake and fiber intake as additional covariates did not substantially affect the association of chronic insomnia with bile acids (Supplementary Table 6).”*

Lines 666-669: *“Given that dietary cholesterol intake and fiber intake might be potential confounders affecting the relationship between the chronic insomnia and the bile acid pool, we further did a sensitivity analysis by including dietary cholesterol intake and fiber intake as additional covariates in the above model 3.”*

5. The authors conclude, “Our results from two large cohort studies provided timely evidence supporting the effect of chronic insomnia on the gut microbiome”. This conclusion is not supported by the data as it implies directionality. No evidence was presented to suggest that insomnia causes a change in the gut microbiome

when the inverse could also be true.

Response: As suggested, we have now revised the sentence in the revised manuscript to avoid potential confusion about the causality:

Lines 264-265: *“Our results from two large cohort studies provided timely evidence supporting that chronic insomnia was associated with the variation of gut microbiome.”*

6. The connection to insomnia is diluted progressively throughout the manuscript. By Figure 4, the data presented are only tangentially related to insomnia. The manuscript does not flow in a logical progression.

Response: We appreciate this careful assessment about the logic of our manuscript. We will take this opportunity to clarify the logical progression of the present study in details. For figure 4, it is true that we did not directly mention chronic insomnia, however, all the bile acid metabolites or the microbes we examined were identified based on their close relationships with chronic insomnia. These are reasonable follow-up analyses in an epidemiological study to assess the correlates of chronic insomnia-related microbial features.

The logical progression of the manuscript were:

- We first investigated the chronic insomnia-associated gut microbiota and bile acid features.
- We examined the association of the chronic insomnia-related gut microbiota/bile acids with cardiometabolic diseases.
- We then evaluated whether the above identified bile acids could mediate the association of chronic insomnia-related microbiota with cardiometabolic diseases.
- To develop potential prevention strategies, we explored the prospective association

of dietary factors with the above identified chronic insomnia-related microbial and bile acid biomarkers in the study participants.

To make these points clearer, we have modified the related sentences in the revised manuscript:

Lines 173-174: *“The chronic insomnia-related gut microbial features and bile acids were associated with CMD and risk factors.”*

Lines 175-177: *“To further investigate whether the chronic insomnia-related gut microbiota or bile acids play a role on CMD, we used multivariable logistic regression to examine the association of the chronic insomnia-related gut microbiota or bile acids with CMD.”*

Lines 226-229: *“We used multivariable linear regression models to investigate the longitudinal association of dietary factors with the chronic insomnia-related microbial and bile acid biomarkers in the GNHS participants without chronic insomnia or CMD at baseline.”*

7. Abstract: The authors state “Chronic insomnia is the second most prevalent mental disorder...” It is understood that insomnia is associated with other mental disorders in some cases, but it seems incorrect to insomniacs themselves a mental disorder.

Response: We agree with you that it may be not appropriate to say that chronic insomnia itself a mental disorder. As suggested, we have revised the related sentences in the manuscript:

Line 36: *“Chronic insomnia is a common sleep disorder, with no effective treatment available.”*

Lines 53-54: “Chronic insomnia is a common sleep disorder with a current estimated global prevalence rate of approximately 10-20%¹⁻³.”

8. For all alpha diversity figures, the text mentions that the figure is Shannon and Chao1 indices, but it remains unclear which one is actually shown. What about ACE and Simpson indices?

Response: Sorry for the potential confusion caused. Now we have clarified these raised issues in the revised manuscript. We examined different alpha diversity parameters, including Observed species, Shannon and Chao1. As an exemplar and to make the main results more concise, we only presented results of one alpha diversity parameter (i.e., Observed species) in the main figures: Figure 1 and Figure 2. Results of Shannon and Chao1 indices were presented in the Extended Data Fig. 1-3. We did not examine ACE or Simpson indices previously, but now, as suggested, we further investigated the association of chronic insomnia status with ACE index and Simpson index and added the results to the Extended Data Fig. 1-3.

We have modified the related sentences in the revised manuscript:

Lines 111-118: “The α -diversity parameters (Observed species, Chao 1 index and ACE index) of the New-onset group and Long-term chronic insomnia group were significantly lower compared with those of the Long-term healthy group (Fig. 1b and Extended Data Fig. 1). The α -diversity parameter (Shannon index) of the Long-term chronic insomnia group was significantly lower compared with that of the Long-term healthy group (Extended Data Fig. 1). The α -diversity parameter Simpson index was not significantly different among the four chronic insomnia status groups (Extended Data Fig. 1).”

Lines 125-128: “In the GNHS, chronic insomnia was associated with lower levels of Observed species ($p < 0.01$), Shannon index ($p < 0.05$), Chao 1 index ($p < 0.001$) and

ACE index ($p < 0.001$), respectively (Fig. 2a and Extended Data Fig. 2)."

Figure legends: Lines 790-794: *"The association of chronic insomnia with overall microbiome α -diversity parameter Observed species was evaluated using a multivariable linear regression, adjusted for potential confounding factors (three models in the text). The results of Shannon index, Chao 1 index, ACE index and Simpson index are reported in Extended Data Fig.1."*

Figure legends: Lines 804-806: *"The results of Shannon index, Chao 1 index, ACE index and Simpson index are reported in Extended Data Fig. 2 (discovery cohort) and Extended Data Fig. 3 (validation cohort)."*

9. Figure 2 a-c and e, exact p/q values should be shown instead of asterisk. This should be applied to entire manuscript.

Response: As suggested, we have now shown the exact p/q -value in the Figure 2. Meanwhile, we applied the same rational to all the other figures. Please see our revised figures (Figures 1-4 and Extended Data Figure 1-7).

10. The authors report that Ruminococcaceae UCG-002 and Ruminococcaceae UCG-003 are significantly negatively associated with insomnia. The authors perform subsequent correlations and report various conclusions under this assumption. However, these data are based on the very liberal cutoff of * $q < 0.25$ in Fig. 2c and Supplementary Table 3. The cutoff of * $q < 0.25$ in Fig. 2c and Supplementary Table 3 is not appropriate. This indicates that one quarter of values indicated as significant will be false positive. Maximum of $q < 0.05$ should be used.

Response: Thank you for the suggestion. We agree with you that we should be careful about the potential false positive results. FDR was used to correct for multiple

comparison, as accepted by many other studies, and it can be set in different rates (from 0.05 to 0.25) (Korem, et al, *Cell Metabolism*, 2017, 25:1243-1253.e5; Wang, et al, *Nature Medicine*, 2021, 27:333–43; Asnicar, et al, *Nature Medicine*, 2021, 27:321–32), which allow different levels of discovery rates. FDR (q) <0.25 is commonly used in the microbiome field for the feature selection in the MaAsLin analysis (Wang, et al, *Nature Medicine*, 2021, 27:333–43; Morgan, et al, *Genome biology*, 2012, 13(9), R79).

The q -values can serve as an exploratory guide and the cutoff of q value < 0.25 is provided to be reasonable and acceptable (Jones, et al, *Journal of clinical epidemiology*, 2008, 61(3), 232–240).

More importantly, whether we use the q -value of 0.05 or 0.25 as the cutoff, the results can still be false positive. A best way to confirm the findings is to replicate it in an independent study. In our case, we replicated the chronic insomnia - gut microbe association in an independent cohort involving >6000 participants. Taken together, we can confidently confirm that our discovered chronic insomnia-related microbial features should be reliable.

We have cited the related articles in the revised manuscript:

Lines 655-659: “We used *Multivariate Analysis by Linear Models (MaAsLin)* to identify potential chronic insomnia associated gut microbiota (q value < 0.25 was used as the threshold of significance in the exploratory analyses, as commonly used previously^{26,79}) using the above three different statistical models by comparing the Chronic insomnia group with the Long-term healthy group.”

11. Figure 2e—Insomnia group is shown before control, this should be reversed.

Response: Done (new Figure 2e).

12. Figure 3a and b—the association with stroke is not significant? This is why exact p values need to be shown.

Response: As suggested, now we added exact p values into the figures, including figure 3a and b. In addition, to make the results clearer, we added the corresponding 95% confidence intervals.

We have modified the related figure legends in the revised manuscript:

Figure legends: Lines 847-848: *“Values presented are odds ratio (95%confidence intervals) with corresponding p-values.”*

Figure legends: Lines 850-851: *“Values presented are odds ratio (95%confidence intervals) with corresponding p-values.”*

13. Figure 3f—this is not an appropriate use of a Sankey diagram. The purpose of a Sankey is to show proportionality of flow.

Response: Thank you for the suggestion. As suggested, we replaced the Sankey diagram with a new parallel coordinates chart in the Figure 3f in the revised manuscript.

We have modified the related figure legends in the revised manuscript:

Figure legends: Lines 858-863: *“Parallel coordinates chart showing the association among gut microbes, bile acid biomarkers and CMD outcomes. The left panel shows*

the microbial biomarkers, the middle panel shows the bile acid biomarkers, and the right panel shows the CMD outcomes. The red lines across panels indicate the positive association. The green lines across panels indicate the inverse association.”

14. The conclusion that tea consumption was significantly associated with increased levels of R. UCG-002 is interesting but the authors failed to state whether or not tea consumption was associated with lower levels of insomnia.

Response: As suggested, we further investigated the association of tea consumption with chronic insomnia in the GNHS and GGMP, and used meta-analysis to pool the effect estimate from the GNHS and the GGMP. Meta-analysis of results from the two cohorts showed that the tea consumption was significantly inversely associated with chronic insomnia (Pooled OR: 0.72, 95%CI: 0.55–0.95, $p=0.020$).

We have added the results in the Supplementary Table 11 and in the revised manuscript:

Lines 240-243: *“In addition, meta-analysis of results of the tea consumption-chronic insomnia association from the two cohorts showed that tea consumption (yes versus no) was inversely associated with risk of chronic insomnia (Pooled OR: 0.72, 95%CI: 0.55–0.95, $p=0.020$) (Supplementary Table 11).”*

Lines 746-749: *“In addition, we further investigated the association of tea consumption with risk of chronic insomnia using logistic regression in the GNHS and GGMP, and used random effects meta-analysis to pool the effect estimates from the GNHS and GGMP.”*

15. This manuscript would require significant English language revisions.

Response: We have carefully gone through the manuscript and revised the English language as suggested.

Reviewer #2:

Summary: This research studies whether gut microbiota plays a role in the association between chronic insomnia and cardiometabolic diseases (CMD). Two microbial mediators are identified in a longitudinal cohort and replicated in another independent cohort. Mediator role of bile acids for the association between gut microbiota and CMD is examined using bi-directional mediation analysis. Microbiota-bile acid axis may be a potential intervention target to diminish the impact of chronic insomnia on cardiometabolic health. This manuscript is well organized and well written. The study groups involved are large while one group is used as replication. I have some minor comments as follows.

Response: We thank you for these positive comments.

1. Line 102: why "In the GGMP, participants were divided into two groups: (i) Non-chronic insomnia group, and (ii) Chronic insomnia group (Methods)" while in the GNHS participants were divided into four?

Response: In the GNHS, we have assessed the chronic insomnia status twice: at baseline and at follow-up. At each time point, we can divide them into two groups. Given these longitudinal data, we can characterize the trajectory of chronic insomnia status and divided them into four groups as we stated in our original manuscript (lines 97-103): "(i) Long-term healthy group (i.e., without chronic insomnia at baseline or follow-up), (ii) Recovery group (i.e., from chronic insomnia at baseline to normal at follow-up), (iii) New-onset group (i.e., without chronic insomnia at baseline but with chronic insomnia at follow-up), (iv) Long-term chronic insomnia group (i.e., with chronic insomnia at baseline and follow-up)".

In the GGMP, we assessed the chronic insomnia status only once given its cross-

sectional study design, therefore, we can only divided participants into two groups. We have revised the sentence in the text to clarify the study design. (Lines 103-105)

Lines 103-105: “*In the GGMP, given the cross-sectional study design, participants were divided into two groups: (i) Non-chronic insomnia group, and (ii) Chronic insomnia group (Methods).*”

2. Line 128: Is "Multivariate Analysis by Linear Models (MaAsLin)" the same as a linear regression with multiple predictors? Is chronic insomnia the outcome variable? If so, shouldn't logistic regression be used given chronic insomnia is dichotomous?

Response: In MaAsLin, chronic insomnia is the exposure (predictor) variable while the different microbiota (arcsin-square root transformed abundance) are the outcome variables. Therefore, it is not the same as linear regression with multiple predictors and logistic regression is not appropriate. We have now clarified in the text:

Lines 136-138, “*We used Multivariate Analysis by Linear Models (MaAsLin) to identify potential microbial biomarkers (as outcome variables) of chronic insomnia (as a predictor in the model), adjusted for potential confounders.*”

3. Lines 139-150: What are the results obtained from the "orthogonal partial least squares discrimination analysis (OPLS-DA)"? From which method (OPLS-DA or logistic regression) these p -values are?

Response: We used OPLS-DA method to screen the potential fecal bile acids associated with chronic insomnia. Therefore, the results of OPLS-DA are 10 bile acids, which were presented in the *Extended Data Fig. 5*. Based on the initial findings from the OPLS-DA method, we further used the linear regression model (sorry for the typo in the results section, we have now corrected it. In the method section, we clearly

said that we used the linear regression to do the analysis) to identify the potential bile acids associated with chronic insomnia, adjusted for the potential covariates. The results are presented in the Fig. 2e. Therefore, the p -values are derived from linear regression models.

We have modified the related sentences in the revised manuscript:

Lines 155-158: *“We next used orthogonal partial least squares discrimination analysis (OPLS-DA) to identify potential fecal bile acids associated with chronic insomnia (Extended Data Fig. 5) and then used the linear regression to confirm the chronic insomnia-bile acid associations in the GNHS (Fig. 2e).”*

Lines 662-666: *“Next, we used orthogonal partial least squares discrimination analysis (OPLS-DA) to identify potential bile acids associated with chronic insomnia. For those OPLS-DA selected bile acids, we further used linear regression to confirm the association of chronic insomnia with these bile acids, adjusted for the same covariates as above model 3.”*

4. Line 152: In this section it would be nice to mention the statistical method used, like in the previous sections.

Response: Thank you for the suggestion. As suggested, we have added the related sentences in the revised manuscript:

Lines 175-177: *“To further investigate whether the chronic insomnia-related gut microbiota or bile acids play a role on CMD, we used multivariable logistic regression to examine the association of the chronic insomnia-related gut microbiota or bile acids with CMD.”*

5. Line 215: change "relate" to "related"?

Response: Done (Line 253).

Reviewer #3:

Summary: This is an interesting and important study to examine the associations of insomnia with gut microbiome and its role in cardiometabolic disease risk. Overall, the study is well designed and sufficiently powered, and the analysis appears appropriate. Below are my comments/suggestions to further improve the manuscript.

1. Introduction: the authors need to provide stronger rationales for their study in the introduction. For example, the concept of the brain-gut axis should be mentioned. Also, 1-2 sentences could be added to summarize prior studies showing that gut microbiomes also exhibit circadian rhythms, which interact with the host circadian rhythms, and sleep disturbances such as chronic insomnia could in turn disrupt microbial circadian rhythms, thus influencing gut microbial composition and activity.

Response: We thank you for your suggestions. As suggested, we have revised the manuscript and added more related backgrounds in the introduction section:

Lines 63-68: *“The gut microbiome is vital to the human health^{10,11}. Of note, the brain-gut axis has been intensively studied in the past few years¹²⁻¹⁴. Prior studies have reported that the gut microbiome exhibited circadian rhythms, which showed interaction with the host circadian rhythms¹⁵⁻¹⁷. Sleep disturbances, such as chronic insomnia, could in turn disrupt microbial circadian rhythms, thus influencing gut microbial composition and function¹⁸⁻²².”*

-
2. It seems that the definition and assessment of chronic insomnia were somewhat different between the discovery and validation cohort. What impact do the authors think this difference may have on the results?

Response: It is true that the definition and assessment of chronic insomnia were different, due to the slightly different questionnaires. However, we would like to argue that the difference in the definition is minor and the impact on the results is minimal. In the GNHS, we asked whether they have chronic insomnia for at least 3 days a week for at least six months, while in the GGMP, it was for at least 3 days a week for at least one month. Nevertheless, the best way to test the difference is to compare the results between the two cohorts. Here, our results in the GNHS cohort (as discovery cohort, using a very strict definition of chronic insomnia) were successfully replicated in the GGMP cohort. Thus, although we cannot tell the exact impact of the two definitions on the whole gut microbiome, our results about the chronic insomnia-related gut microbes identified from the GNHS should be reliable. In any way, we have added a sentence in the manuscript to discuss the potential impact of the definition.

Lines 335-338: “Third,, and the potential impact of the two slightly different definitions of chronic insomnia between the GNHS and GGMP is still unclear, although our results are successfully replicated.”

3. Another common way to assess insomnia is to use a symptom score to characterize the severity of the disease. The authors could conduct a secondary analysis evaluating the number of insomnia symptoms in relation to the gut microbial taxa/metabolites identified in the primary analysis. If a dose-response relationship is found, this could provide further support to their study conclusion.

Response: Thank you for the suggestion. We totally agree with this reviewer’s suggestion about the secondary analysis. In the GNHS, we were not able to calculate the chronic insomnia symptom score to conduct the suggested secondary analysis. Nevertheless, in the GGMP, we calculated the chronic insomnia symptom score. As

suggested, we conduct the secondary analysis to evaluate the number of insomnia symptoms in relation to the gut microbial taxa identified in the primary analysis. Multivariable linear regression was used to assess the association of chronic insomnia symptom score with gut microbial biomarkers, adjusted for the same covariates. The secondary analysis results supported our study conclusion.

We have added the results in the Supplementary Table 5 and in the revised manuscript:

Lines 148-153: *“To assess the potential influence of the number of insomnia symptoms, we conducted a secondary analysis in the GGMP, where the data were available, and found that per unit change in the chronic insomnia symptom score was inversely associated with per 1-SD change in Ruminococcacea UCG-002 (β : -0.04, 95%CI: -0.06--0.02, $p < 0.001$) and Ruminococcacea UCG-003 (β : -0.04, 95%CI: -0.07--0.01, $p = 0.002$) (Supplementary Table 5).”*

Lines 722-724: *“We conducted a secondary analysis to evaluate the association of the insomnia symptom score (per unit change) with the identified gut microbiota biomarkers by using the linear regression, adjusted for the same covariates.”*

4. The authors claimed that “tea consumption may alleviate the detrimental impact of chronic insomnia on the cardiometabolic health”. However, this suggestion was not directly supported by the results presented. The authors can perform a stratified analysis by tea consumption to examine whether the associations between insomnia-related gut microbiomes and CMD risk factors are weaker among participants with high habitual tea consumption versus those with low consumption.

Response: As suggested, we performed a stratified analysis by tea consumption in the GNHS. The detailed results are shown in the Supplementary Table 10. The results of the inverse association between *Ruminococcacea UCG-002* and CMD risk factors (especially for diabetes and dyslipidemia) are in general stronger among those with habitual high tea consumption versus low tea consumption groups. These results

support that tea intake may alleviate the detrimental impact of chronic insomnia on the CMD risk via increasing *Ruminococcaceae* UCG-002 (because this microbe is beneficial for CMD and chronic insomnia may decrease it). We added these stratified analyses into the manuscript. Meanwhile, we have thoroughly revised our manuscript to town done our claims about the causality given the observational nature of the cohort study.

Lines 235-240: *“Furthermore, the stratified analysis by tea consumption (yes versus no) in the GNHS showed that the inverse association between Ruminococcaceae UCG-002 and CMD risk factors (especially for T2D (OR: 0.73, 95%CI: 0.60–0.89, p = 0.002) and dyslipidemia (OR: 0.85, 95%CI: 0.74–0.98, p = 0.024)) were in general stronger among those with habitual tea consumption (Supplementary Table 10).”*

Lines 743-746: *“We also performed the additional stratified analyses by tea consumption (yes versus no) using logistic regression in the GNHS to explore whether the associations between chronic insomnia-related gut microbiome and CMD risk factors could be affected by tea consumption.”*

5. For the primary findings, additional stratified analyses (at least by age and sex) are needed to explore potential heterogeneity in the association (it is known that both insomnia and CMD risk are strongly related to age and sex).

Response: Thank you for the suggestion. As suggested, we performed additional interaction analysis by age and sex to explore potential heterogeneity for the chronic insomnia-gut microbiota association and gut microbiota-CMD association in the GNHS and GGMP, and used random effects meta-analysis to pool the effect estimates from the GNHS and the GGMP. We presented the results of stratification analyses in the main text if a significant interaction (potential heterogeneity) was found, and put other stratified results of non-significant interaction in the supplemental tables.

We have added the results in the Supplementary Table 4 and 8 and added the related

sentences in the revised manuscript:

Lines 146-148: “*In addition, chronic insomnia had no interactions with age or sex for Ruminococcacea UCG-002 and Ruminococcacea UCG-003 (Supplementary Table 4).*”

Lines 197-202: “*Ruminococcacea UCG-002 was interacted with sex on the risk of dyslipidaemia ($p_{\text{interaction}} = 0.003$) (Supplementary Table 8). The stratified analyses by sex showed that the inverse association of Ruminococcacea UCG-002 with dyslipidaemia was significant among men participants (Pooled OR: 0.87, 95%CI: 0.81–0.93, $p < 0.001$), but not among women participants (Pooled OR: 0.96, 95%CI: 0.88–1.05, $p = 0.394$) (Supplementary Table 8).*”

Lines 733-736: “*In addition, we further performed interaction analysis and additional stratified analyses by age and sex to explore potential heterogeneity for the chronic insomnia-gut microbiota association and gut microbiota-CMD association, and used random effects meta-analysis to pool the effect estimates from the GNHS and GGMP.*”

6. Figure 1 can be improved by clearly indicating when stool samples were collected, when insomnia assessment was done and repeated (two time points), and when cardiometabolic risk factors were assessed. Such information can help readers better understand the study design.

Response: Done (Figure 1a).

7. Supplemental Table 2: I would suggest presenting the sample characteristics by chronic insomnia status in the validation cohort too, similar to the discovery cohort. It would be helpful to compare the prevalence of chronic insomnia and understand any potential differences between the two study cohorts.

Response: As suggested, we have now added characteristics by chronic insomnia status in the validation cohort in the Supplemental Table 2.

8. Discussion: The findings on secondary bile acids are very interesting. Some prior studies also support these findings, but the authors did not include them in their discussion. For example, in one study that examined insomnia symptoms and plasma metabolomics (PMID: 30371783), derivatives of second bile acids (e.g., glyoursodeoxycholic acid) were positively associated with more severe insomnia symptoms. Another study found that Ruminococcaceae was related to plasma LDL and triglycerides, which has direct implications for cardiovascular risk (PMID: 31862950).

Response: Thank you for these helpful suggestions. As suggested, we now cited these two important papers and discussed their findings in the discussion section of the revised manuscript as followed:

Lines 285-294: *“The results were consistent with several recent studies^{41,52-55}. One study indicated that secondary bile acid metabolites (i.e., glyoursodeoxycholate) might link poor habitual sleep quality and coronary heart disease risk⁵². Another study showed that Ruminococcaceae UCG-002 was positively associated with insulin sensitivity in patients with polycystic ovary syndrome⁴¹. In addition, another recent study demonstrated that Ruminococcaceae UCG-002 and Ruminococcaceae UCG-003 were positively associated with several plasma HDL subclasses, and were inversely associated with several plasma LDL subclasses, which had direct beneficial implications for cardiovascular health⁵³.”*

9. Discussion: Sleep-disordered breathing often leads to chronic insomnia and poor sleep quality. Intermittent hypoxia from sleep-disordered breathing may also have an impact on gut microbiome, as suggested by some recent evidence (PMID: 33705556,29896566, 26711739). As sleep-disordered breathing was not considered in the current study, the authors should discuss this and acknowledge this as a limitation (or a potential explanation to their findings)

Response: As suggested, we have added the discussion in the text and acknowledged this as a limitation in the revised manuscript:

Lines 338-341: *“Fourth, we do not collect information on sleep-disordered breathing, which is closely associated with chronic insomnia and may have an impact on gut microbiome and bile acid metabolism⁷⁰⁻⁷².”*

10. Discussion: Given that caffeine in tea may exacerbate insomnia, the authors need to be cautious when interpreting tea consumption as a potential intervention strategy to reduce the adverse cardiometabolic impact of insomnia-related gut microbial alterations.

Response: We agree with you. As suggested, we have revised the related sentences in the discussion to avoid potential confusion caused by our previous causal language (lines 311-313, lines 315-320 and lines 347-348). We also discussed the potential influence of caffeine in tea in the discussion (lines 320-323).

Lines 311-313: *“We found that habitual tea consumption was prospectively associated with the identified gut microbiota and bile acids in an opposite direction compared with chronic insomnia.”*

Lines 315-320: *“Mechanism underlies the association of habitual tea consumption with the gut microbiota-bile acid axis may be attributed to their rich contents of tea polyphenols, flavonoids, alkaloids, and various antioxidant compounds, which are reported to have the ability to modulate the gut microbial composition and bile acid metabolism⁶⁴⁻⁶⁶, and improve the circadian rhythm system presented in the brain and gut^{67,68}.”*

Lines 347-348: *“Habitual tea consumption had an inverse association with the chronic insomnia-disrupted gut microbiota and bile acids.”*

Lines 320-323: *“Nevertheless, we can’t establish a causal relationship between the tea consumption and CMD-related gut microbiota at this stage and these above speculations should be explained with caution, especially given the fact that caffeine in tea may exacerbate insomnia⁶⁹.”*

Reviewer #4:

Summary: The authors investigated the associations between chronic insomnia, bile acid, and CMD. The authors performed a mediation analysis to uncover plausible mechanisms of microbial effects on CMD through specific bile acids. I commend the authors on applying mediation analysis to investigate mechanisms of chronic insomnia on CMD, however their analyses fall short on a few important points mentioned below. Overall, the authors need a more careful consideration of the causal assumptions for mediation analysis and clearer reporting of the methods used.

Response: We thank you for the suggestions about the mediation analysis and causal assumption. Now, we have clearly reported the related results as suggested. The detailed responses to your comments are shown below:

1. It is unclear what the authors mean with the term “bi-directional” mediation analysis. This is not a commonly used term for mediation analysis. Similarly, the term “inverse” mediation is not commonly used.

Response: Thank you for the suggestion. As indicated, bi-directional mediation analysis is not a commonly used word. To avoid potential confusion caused by this word and related “inverse” mediation, we have deleted the term “bi-directional” in the revised manuscript and only keep traditional one-direction mediation analysis (Line

45, Line 211 and Line 692).

We have also revised the sentences in the figure legends:

Figure legends: Lines 867-870: *“The gray lines indicate the associations, with corresponding normalized beta values and p values. The red arrowed lines indicate the microbial effects on CMD mediated by specific bile acid biomarkers, with the corresponding mediation p values. p value < 0.05 is significantly different.”*

2. Most importantly, it appears the authors tried to demonstrate evidence for mediation of microbial effects -> bile acid -> CMD as opposed to mediation microbial effects -> CMD -> bile acid. There are two major limitations to this. First, there are six different plausible models of the three variables – the authors only investigated two plausible models. Second, and the most important, comparing the statistical significance and/or magnitude of a mediated effect estimate from one model (e.g., microbial effects -> bile acid -> CMD) to the statistical significance and/or magnitude of a mediated effect estimate from another model (e.g., microbial effects -> CMD -> bile acid) does not provide evidence for mediation through one mechanism over the other. These models are from the same equivalence class and therefore cannot be distinguished from one another by statistical tests alone. There must be a compelling scientific reason to suspect that these are the two plausible mediation models and even if there is compelling scientific rationale for these two models, statistics cannot distinguish between these two models. Please see Thoemmes (2015) for more details.

Thoemmes, F. (2015). Reversing arrows in mediation models does not distinguish plausible models. *Basic and Applied Social Psychology*, 37(4), 226–234. <https://doi.org/10.1080/01973533.2015.1049351>

Response: We thank you for these comments, which we found really helpful. We have carefully read the literature you provided on mediation analysis. We again read

related literature about gut microbiota and bile acids (*Thomas, et al, Nature reviews. Drug discovery, 2008, 7(8), 678–693*; *Schaap, et al. Nature reviews. Gastroenterology & hepatology, 2014, 11(1), 55–67*; *Song, et al, Microbiome, 2019, 7(1), 9*; *Pi, et al, mSystems, 2020, 5(3), e00176-20*; *Taxonomic identification of BSHs in HMP database*). Based on the available evidence, we hypothesized that bile acid biomarkers of chronic insomnia mediated the association of microbial biomarkers of chronic insomnia with cardiometabolic diseases (mediation of microbial effects -> bile acids -> CMD) based on the biological causalities and our findings:

1) In the present study, we demonstrated the trajectory of chronic insomnia status was inversely associated with *Ruminococcaceae UCG-002* and *Ruminococcaceae UCG-003* based on the longitudinal data of chronic insomnia status over the past 6 years. (chronic insomnia -> gut microbiota)

2) In the longitudinal analysis among GNHS, we demonstrated *Ruminococcaceae UCG-002* and *Ruminococcaceae UCG-003* were inversely associated with the cardiometabolic disease risk factor (dyslipidemia). (gut microbiota -> CMD)

3) Gut microbiota, but not humans, have the ability to convert primary bile acids to secondary bile acids (*Witkowski, et al, Circulation research, 2020, 127(4), 553–570*). Previous studies have demonstrated that the family *Ruminococcaceae* harbors BSH and 7 α -dehydroxylase activity, which can convert primary bile acids into secondary bile acids (*Song et al, Microbiome, 2019, 7(1), 9*; *Pi et al, mSystems, 2020, 5(3), e00176-20*; *Taxonomic identification of BSHs in HMP database*). *Ruminococcaceae UCG-002* and *Ruminococcaceae UCG-003* are belonged to the family *Ruminococcaceae*, which can convert primary bile acids into secondary bile acids. (gut microbiota -> bile acids)

4) Moreover, *Ruminococcaceae UCG-002* and *Ruminococcaceae UCG-003* in the present study were positively associated with secondary bile acids and were inversely associated with primary bile acids. (gut microbiota -> bile acids)

5) Several studies suggested that treatment with specific microbial derived secondary bile acids (obeticholic acid, deoxycholic acid and glycodeoxycholic acid) in patients with T2D could improve insulin sensitivity and HbA1c (*Mudaliar, et al, Gastroenterology, 2013, 145, 574-582 e571*). (bile acids -> CMD)

6) Previous studies also demonstrated that administration of specific bile acids (e.g. hyocholic acid and hyodeoxycholic acid) in mice improved glucose homeostasis through activation of their receptor Farnesoid-X receptor (FXR) and inhibition of their receptor Takeda-G-protein-receptor-5 (TGR5) (*Makishima, et al, Science, 1999, 284, 1362-1365; Wang, et al, Mol Cell, 1999, 3, 543-553; Kawamata, et al, 2003, J Biol Chem, 278, 9435-9440; Zheng, Cell metabolism, 2021, 33(4), 791-803.e7*). (bile acids -> CMD)

Therefore, based on the above evidence and the hypothesis, we only do the mediation analysis for one model: microbial effects -> bile acid -> CMD.

We have added the sentence in the revised manuscript:

Line 691-694: *“Based on the biological plausibility for the associations among gut microbiota, bile acids and CMD^{54,59,80,81}, and our above findings, we performed the mediation analysis to evaluate whether bile acids could mediate the association of chronic insomnia related-gut microbiota with CMD outcomes (gut microbiota → bile acids → CMD).”*

Line 302-306: *“Ruminococcaceae UCG-002 and Ruminococcaceae UCG-003 may have the ability to convert some primary bile acids into secondary bile acids as they belonged to the bile salt hydrolase (BSH) and 7 α -dehydroxylase-active family Ruminococcaceae, which harbors many secondary bile acid-producing genera such as Faecalibacterium and Ruminniclostridium^{58,59}.”*

3. It is unclear how the authors estimated the mediated effects. There are many statistical methods used in the mediation literature to estimate mediated effects and perform significance testing of mediated effects and it is important that the authors clarify this. Additionally, the authors did not mention the necessary causal assumptions that are required when performing mediation analysis. The interpretation of mediated effects relies on specific no unmeasured confounding

assumptions.

MacKinnon, D. P. (2008). Introduction to statistical mediation analysis. Routledge.

Vanderweele, T. J. (2015). Explanation in Causal Inference: Methods for Mediation and Interaction (Vol. 53). <https://doi.org/10.1017/CBO9781107415324.004>.

Valente, M. J., Rijnhart, J. J. M., Smyth, H. L., Muniz, F. B., & Mackinnon, D. P. (2020). Causal Mediation Programs in R , Mplus , SAS , SPSS , and Stata. Structural Equation Modeling: A Multidisciplinary Journal, 00(00), 1–10. <https://doi.org/10.1080/10705511.2020.1777133>

Response: As suggested, we have now clarified the mediation method in the revised manuscript:

Lines 695-708: *“The mediation analysis was performed to examine the mediating effect of bile acids in the association of chronic insomnia-related gut microbiota with CMD outcomes⁸². We defined three pathways in the mediation analysis: (1) exposure to mediator; (2) mediator to outcome; (3) exposure to outcome. In the mediation analysis, the covariates included: age, sex, BMI, smoking status, alcohol status, physical activity, education, income and total energy intake. The mediation analysis was performed using the R-mediation package with same parameter settings (boot = “TRUE”, boot.ci.type = “perc”, conf.level = 0.95, sims=1000). The total effect was obtained through the sum of a direct effect and a mediated (indirect) effect. Percentage of the mediated effect was calculated using the formula: (mediated effect/total effect) × 100. The sensitivity analysis was performed to test the robustness of the mediation effect and violation of the assumption (sequential ignorability) using R-medsens package with default parameters^{83,84}. The reporting of mediation results followed the Guideline for Reporting Mediation Analyses (AGReMA) statement⁸⁵.”*

As we responded to your second comment, we have a strong biological rationale and hypothesis to do the mediation analysis. We have mentioned the causal assumptions in the revised manuscript:

Introduction section Lines 68-74: *“On the other hand, the gut microbial dysbiosis is associated with the development of CMD, and has a substantial impact on the metabolic health²³⁻²⁹. Meanwhile, the dysregulation of bile acid metabolism and its interaction with gut microbiome are also closely associated with host metabolic health³⁰⁻³³. Repeated sleep disruption in mice has led to a persistent change in gut microbiota composition and changes in bile acid metabolism³⁴⁻³⁶. Therefore, we hypothesized that the gut microbiota-bile acid axis may play a role in linking chronic insomnia and CMD.”*

Method section Lines 691-694: *“Based on the biological plausibility for the associations among gut microbiota, bile acids and CMD^{54,59,80,81}, and our above findings, we performed the mediation analysis to evaluate whether bile acids could mediate the association of chronic insomnia related-gut microbiota with CMD outcomes (gut microbiota → bile acids → CMD).”*

4. There are also sensitivity analyses that are designed to assess the robustness of mediated effects to violations of these assumptions. For example of a sensitivity analysis method see:

Cox, M. G., Kisbu-Sakarya, Y., Miocevic, M., & MacKinnon, D. P. (2013). Sensitivity Plots for Confounder Bias in the Single Mediator Model. *Evaluation Review*, 37(5), 405–431. <https://doi.org/10.1177/0193841X14524576>.

Imai, K., Keele, L., & Yamamoto, T. (2010). Identification, Inference and Sensitivity Analysis for Causal Mediation Effects. *Statistical Science*, 25(1), 51–71. <https://doi.org/10.1214/10-STS321>.

Smith, L. H., & Vanderweele, T. J. (2019). Mediation E-values: Approximate

sensitivity analysis for unmeasured mediator-outcome confounding. *Epidemiology*, 30(6), 835–837. <https://doi.org/10.1097/EDE.0000000000001064>.

Vanderweele, T. J. (2010). Bias formulas for sensitivity analysis for direct and indirect effects. *Epidemiology*, 21(4), 540–551. <https://doi.org/10.1097/EDE.0b013e3181df191c>.

Response: As suggested, we went through the references you provided and then, using the method mentioned in the reference, we performed the sensitivity analysis of mediation with the R-medsens package.

We have added the results in the supplementary table 9 and in the revised manuscript:

Lines 221-223: “*Sensitivity analysis for mediation effects indicated that the results of the above mediation analysis were relatively robust to the possible existence of an unmeasured confounder (Supplementary Table 9).*”

Lines 705-707: “*The sensitivity analysis was performed to test the robustness of the mediation effect and violation of the assumption (sequential ignorability) using R-medsens package with default parameters^{83,84}.*”

5. The authors would be encouraged to follow the AGRReMA statement for the reporting of mediation results. The following reference contains the necessary reporting guidelines for mediation analysis and many key citations that I would highly encourage the authors consult.

Lee, H., Cashin, A. G., Lamb, S. E., Hopewell, S., Vansteelandt, S., VanderWeele, T. J., ... & Henschke, N. (2021). A Guideline for Reporting Mediation Analyses of Randomized Trials and Observational Studies: The AGRReMA Statement. *JAMA*,

326(11), 1045-1056.

Response: Thank you for the suggestion. As suggested, we reported the mediation results following the AGR_eMA statement and also cited the mentioned reference.

We have added related information in the revised manuscript:

Method section Lines 707-708: “*The reporting of mediation results followed the Guideline for Reporting Mediation Analyses (AGR_eMA) statement⁸⁵.*”

Dear reviewers,

RE NCOMMS-21-28734-T

Chronic insomnia, gut microbiota-bile acid axis and cardiometabolic health: results from two large-scale human cohorts

We would like to thank you for your helpful comments and suggestions for improving our manuscript. We have uploaded a clean version of the revised manuscript and a version with changes highlighted in yellow colour. Our point-by-point responses to your comments are presented below.

Reviewer #1:

Summary: In this manuscript, Jiang, Zhuo, and He et al. observed in two large, independent human studies that subjects suffering from insomnia had different gut microbiomes than healthy controls. Furthermore, the bile acid pool and risk for CMD were altered in insomniacs. While the size of the human sample pool is impressive, the conclusions in this manuscript are repeatedly overstated and, in several instances, are not supported by the data. This manuscript lacks any causative data which dampens overall enthusiasm for publication. Specific recommendations are outlined below.

Response: We thank you for your helpful comments. We took your suggestions and modified our conclusions to tone down our claims to avoid potential overstatement. We also revised the related statements and discussion about the causality throughout the text. Please see our responses to your specific comments below:

1. There is a general oversimplification linking alterations in gut microbiome composition to alterations in bile acids (i.e. microbiome = bile acids). However, alterations in gut microbial communities will alter many metabolites beyond bile

acids. Secondary bile acids are one of likely thousands of small molecule metabolites produced by commensal bacteria, and a more balanced discussion and investigation is needed of other microbe-associated metabolic pathways.

Response: We agree with you that there are much more metabolites in addition to bile acids in the gut. In our prior version of the manuscript, we mainly focused on the analysis of bile acids as a pre-defined strategy. This is because bile acids are very important biologically active metabolites closely related with both gut microbiome and human metabolic health. Therefore, we hypothesized that the gut microbiota-bile acid axis may potentially play a role linking chronic insomnia and cardiometabolic diseases.

Nevertheless, as you pointed out, alterations in gut microbiota would alter many other metabolites. Thus, as a secondary analysis in the revision of the manuscript, we did additional analyses to estimate the association of chronic insomnia with another two important classes of gut microbe-associated metabolites: short chain fatty acids, aromatic amino acids and their derivatives. We found that chronic insomnia was not associated with short chain fatty acids, aromatic amino acids or their derivatives in our cohort.

We have now added the additional results in the Supplementary Table 7 and in the text of the revised manuscript. We also added a more balanced discussion about this point as suggested in the discussion section (lines 273-278):

Lines 168-170: *“In addition, chronic insomnia was not associated with short-chain fatty acids, aromatic amino acids or their derivatives (Supplementary Table 7).”*

Lines 671-675: *“In addition, we tested the association of chronic insomnia with another two important classes of gut microbial metabolites (short-chain fatty acids, aromatic amino acids and their derivatives) by using multivariable linear regression, adjusted for the same covariates as above model 3.”*

Lines 273-278: *“As alterations in gut microbiota would alter many other metabolites, we did a secondary analysis for the association of chronic insomnia with short-chain fatty acids, aromatic amino acids and their derivatives^{46,47}. Although we did not find any significant association for these two classes of metabolites, we could not rule out the possibility that chronic insomnia may be associated with other metabolites, which needs further investigations.”*

2. Although the authors acknowledge this as a limitation in the discussion, the data presented in this manuscript are purely associative and descriptive. No causative evidence is presented—this severely dampens enthusiasm for publication.

Response: With respect, we understand your concern about the “causality” evidence. However, we want to argue that epidemiological study with prospective study design such as ours is very important and critical for the evidence-based medicine and translation of basic research. For a topic such as our present study to link chronic insomnia, gut microbiota and cardiometabolic diseases in human, it is almost impossible to run a randomized control trial due to the ethical issue and feasibility issue, therefore, we have to rely on the prospective study to assess the temporal relationship between chronic insomnia and cardiometabolic diseases and then assess the role of gut microbiota-bile acid axis. From this kind of prospective study, we can generate high-level evidence to infer causality and propose novel mechanism directly in humans. We also agree that using animal model may provide evidence of causality and related mechanism, but there is usually a huge gap between animal research and human clinical research, especially in the field of gut microbiome. There are big differences in the gut microbiome between mice models (even for humanized gnotobiotic mouse model) and humans.

Taken together, we believe that our observation is novel with important clinical implications and translational value. As you noted, we have added the discussion about the causality in our limitation section.

-
3. Associations are incorrectly reported as causation in several instances or conclusions are drawn based purely on correlative analysis.
- o Lines 206-208, 226-230 and 259-262.

Response: As suggested, we have now revised the related sentences to tone down our claims and avoid potential confusion caused by the previous causal language:

Lines 243-246: *“The results indicated that habitual tea consumption was associated with the gut microbiota-bile acid axis, which may potentially underlie the association between chronic insomnia and CMD (Fig. 4c).”*

Lines 264-268: *“Our results from two large cohort studies provided timely evidence supporting that chronic insomnia was associated with the variation of gut microbiome. Specifically, Ruminococcaceae UCG-002 and Ruminococcaceae UCG-003 were identified as the two potential genera inversely associated with chronic insomnia, and these microbes may be associated with host glucose homeostasis and lipid metabolism^{40,41}.”*

Lines 311-313: *“We found that habitual tea consumption was prospectively associated with the identified gut microbiota and bile acids in an opposite direction compared with chronic insomnia.”*

4. The differences reported in the bile acid pool are not striking. Although the authors corrected for several factors including some dietary substrates, more specific corrections for dietary cholesterol intake or dietary fiber intake could possibly mitigate any significance in the bile acid pool between healthy human subjects and those with insomnia.

Response: Thank you for the suggestion. In our previous version, we did not use dietary cholesterol intake or fiber intake as covariates in the statistical models. To

address this reviewer's concerns, we have done a sensitivity analysis to further adjust for dietary cholesterol intake and fiber intake based on our main model 3, and the results showed that adding dietary cholesterol intake and fiber intake into the model did not substantially change the results. The Benjamini-Hochberg method was used to control the false discovery rate (FDR) for multiple testing.

In the GNHS, sensitivity analysis about the inclusion of dietary cholesterol intake and fiber intake as additional covariates in our statistical model showed that chronic insomnia was associated with higher levels of merocholic acid (MCA, β : 0.21, 95%CI: [0.04, 0.38], $p=0.049$) and norcholic acid (NorCA, β : 0.21, 95%CI: [0.04, 0.37], $p=0.042$), and associated with lower levels of isolithocholic acid (IsoLCA, β : -0.26, 95%CI: [-0.43, -0.09], $p=0.030$), lithocholic acid (LCA, β : -0.21, 95%CI: [-0.38, -0.04], $p=0.035$) and ursodeoxycholic acid (UDCA, β : -0.22, 95%CI: [-0.39, -0.05], $p=0.049$).

We have added these new results in the Supplementary Table 6 and in the revised manuscript:

Lines 161-164: *“The results of sensitivity analysis showed that adding dietary cholesterol intake and fiber intake as additional covariates did not substantially affect the association of chronic insomnia with bile acids (Supplementary Table 6).”*

Lines 666-669: *“Given that dietary cholesterol intake and fiber intake might be potential confounders affecting the relationship between the chronic insomnia and the bile acid pool, we further did a sensitivity analysis by including dietary cholesterol intake and fiber intake as additional covariates in the above model 3.”*

5. The authors conclude, “Our results from two large cohort studies provided timely evidence supporting the effect of chronic insomnia on the gut microbiome”. This conclusion is not supported by the data as it implies directionality. No evidence was presented to suggest that insomnia causes a change in the gut microbiome

when the inverse could also be true.

Response: As suggested, we have now revised the sentence in the revised manuscript to avoid potential confusion about the causality:

Lines 264-265: “Our results from two large cohort studies provided timely evidence supporting that chronic insomnia was associated with the variation of gut microbiome.”

6. The connection to insomnia is diluted progressively throughout the manuscript. By Figure 4, the data presented are only tangentially related to insomnia. The manuscript does not flow in a logical progression.

Response: We appreciate this careful assessment about the logic of our manuscript. We will take this opportunity to clarify the logical progression of the present study in details. For figure 4, it is true that we did not directly mention chronic insomnia, however, all the bile acid metabolites or the microbes we examined were identified based on their close relationships with chronic insomnia. These are reasonable follow-up analyses in an epidemiological study to assess the correlates of chronic insomnia-related microbial features.

The logical progression of the manuscript were:

- We first investigated the chronic insomnia-associated gut microbiota and bile acid features.
- We examined the association of the chronic insomnia-related gut microbiota/bile acids with cardiometabolic diseases.
- We then evaluated whether the above identified bile acids could mediate the association of chronic insomnia-related microbiota with cardiometabolic diseases.
- To develop potential prevention strategies, we explored the prospective association

of dietary factors with the above identified chronic insomnia-related microbial and bile acid biomarkers in the study participants.

To make these points clearer, we have modified the related sentences in the revised manuscript:

Lines 173-174: *“The chronic insomnia-related gut microbial features and bile acids were associated with CMD and risk factors.”*

Lines 175-177: *“To further investigate whether the chronic insomnia-related gut microbiota or bile acids play a role on CMD, we used multivariable logistic regression to examine the association of the chronic insomnia-related gut microbiota or bile acids with CMD.”*

Lines 226-229: *“We used multivariable linear regression models to investigate the longitudinal association of dietary factors with the chronic insomnia-related microbial and bile acid biomarkers in the GNHS participants without chronic insomnia or CMD at baseline.”*

7. Abstract: The authors state “Chronic insomnia is the second most prevalent mental disorder...” It is understood that insomnia is associated with other mental disorders in some cases, but it seems incorrect to insomniac itself a mental disorder.

Response: We agree with you that it may be not appropriate to say that chronic insomnia itself a mental disorder. As suggested, we have revised the related sentences in the manuscript:

Line 36: *“Chronic insomnia is a common sleep disorder, with no effective treatment available.”*

Lines 53-54: *“Chronic insomnia is a common sleep disorder with a current estimated global prevalence rate of approximately 10-20%¹⁻³.”*

8. For all alpha diversity figures, the text mentions that the figure is Shannon and Chao1 indices, but it remains unclear which one is actually shown. What about ACE and Simpson indices?

Response: Sorry for the potential confusion caused. Now we have clarified these raised issues in the revised manuscript. We examined different alpha diversity parameters, including Observed species, Shannon and Chao1. As an exemplar and to make the main results more concise, we only presented results of one alpha diversity parameter (i.e., Observed species) in the main figures: Figure 1 and Figure 2. Results of Shannon and Chao1 indices were presented in the Extended Data Fig. 1-3. We did not examine ACE or Simpson indices previously, but now, as suggested, we further investigated the association of chronic insomnia status with ACE index and Simpson index and added the results to the Extended Data Fig. 1-3.

We have modified the related sentences in the revised manuscript:

Lines 111-118: *“The α -diversity parameters (Observed species, Chao 1 index and ACE index) of the New-onset group and Long-term chronic insomnia group were significantly lower compared with those of the Long-term healthy group (Fig. 1b and Extended Data Fig. 1). The α -diversity parameter (Shannon index) of the Long-term chronic insomnia group was significantly lower compared with that of the Long-term healthy group (Extended Data Fig. 1). The α -diversity parameter Simpson index was not significantly different among the four chronic insomnia status groups (Extended Data Fig. 1).”*

Lines 125-128: *“In the GNHS, chronic insomnia was associated with lower levels of Observed species ($p < 0.01$), Shannon index ($p < 0.05$), Chao 1 index ($p < 0.001$) and*

ACE index ($p < 0.001$), respectively (Fig. 2a and Extended Data Fig. 2)."

Figure legends: Lines 790-794: *"The association of chronic insomnia with overall microbiome α -diversity parameter Observed species was evaluated using a multivariable linear regression, adjusted for potential confounding factors (three models in the text). The results of Shannon index, Chao 1 index, ACE index and Simpson index are reported in Extended Data Fig.1."*

Figure legends: Lines 804-806: *"The results of Shannon index, Chao 1 index, ACE index and Simpson index are reported in Extended Data Fig. 2 (discovery cohort) and Extended Data Fig. 3 (validation cohort)."*

9. Figure 2 a-c and e, exact p/q values should be shown instead of asterisk. This should be applied to entire manuscript.

Response: As suggested, we have now shown the exact p/q -value in the Figure 2. Meanwhile, we applied the same rational to all the other figures. Please see our revised figures (Figures 1-4 and Extended Data Figure 1-7).

10. The authors report that Ruminococcaceae UCG-002 and Ruminococcaceae UCG-003 are significantly negatively associated with insomnia. The authors perform subsequent correlations and report various conclusions under this assumption. However, these data are based on the very liberal cutoff of * $q < 0.25$ in Fig. 2c and Supplementary Table 3. The cutoff of * $q < 0.25$ in Fig. 2c and Supplementary Table 3 is not appropriate. This indicates that one quarter of values indicated as significant will be false positive. Maximum of $q < 0.05$ should be used.

Response: Thank you for the suggestion. We agree with you that we should be careful about the potential false positive results. FDR was used to correct for multiple

comparison, as accepted by many other studies, and it can be set in different rates (from 0.05 to 0.25) (Korem, et al, *Cell Metabolism*, 2017, 25:1243-1253.e5; Wang, et al, *Nature Medicine*, 2021, 27:333–43; Asnicar, et al, *Nature Medicine*, 2021, 27:321–32), which allow different levels of discovery rates. FDR (q) <0.25 is commonly used in the microbiome field for the feature selection in the MaAsLin analysis (Wang, et al, *Nature Medicine*, 2021, 27:333–43; Morgan, et al, *Genome biology*, 2012, 13(9), R79).

The q -values can serve as an exploratory guide and the cutoff of q value < 0.25 is provided to be reasonable and acceptable (Jones, et al, *Journal of clinical epidemiology*, 2008, 61(3), 232–240).

More importantly, whether we use the q -value of 0.05 or 0.25 as the cutoff, the results can still be false positive. A best way to confirm the findings is to replicate it in an independent study. In our case, we replicated the chronic insomnia - gut microbe association in an independent cohort involving >6000 participants. Taken together, we can confidently confirm that our discovered chronic insomnia-related microbial features should be reliable.

We have cited the related articles in the revised manuscript:

Lines 655-659: “We used *Multivariate Analysis by Linear Models (MaAsLin)* to identify potential chronic insomnia associated gut microbiota (q value < 0.25 was used as the threshold of significance in the exploratory analyses, as commonly used previously^{26,79}) using the above three different statistical models by comparing the Chronic insomnia group with the Long-term healthy group.”

11. Figure 2e—Insomnia group is shown before control, this should be reversed.

Response: Done (new Figure 2e).

12. Figure 3a and b—the association with stroke is not significant? This is why exact p values need to be shown.

Response: As suggested, now we added exact p values into the figures, including figure 3a and b. In addition, to make the results clearer, we added the corresponding 95% confidence intervals.

We have modified the related figure legends in the revised manuscript:

Figure legends: Lines 847-848: *“Values presented are odds ratio (95%confidence intervals) with corresponding p-values.”*

Figure legends: Lines 850-851: *“Values presented are odds ratio (95%confidence intervals) with corresponding p-values.”*

13. Figure 3f—this is not an appropriate use of a Sankey diagram. The purpose of a Sankey is to show proportionality of flow.

Response: Thank you for the suggestion. As suggested, we replaced the Sankey diagram with a new parallel coordinates chart in the Figure 3f in the revised manuscript.

We have modified the related figure legends in the revised manuscript:

Figure legends: Lines 858-863: *“Parallel coordinates chart showing the association among gut microbes, bile acid biomarkers and CMD outcomes. The left panel shows*

the microbial biomarkers, the middle panel shows the bile acid biomarkers, and the right panel shows the CMD outcomes. The red lines across panels indicate the positive association. The green lines across panels indicate the inverse association.”

14. The conclusion that tea consumption was significantly associated with increased levels of R. UCG-002 is interesting but the authors failed to state whether or not tea consumption was associated with lower levels of insomnia.

Response: As suggested, we further investigated the association of tea consumption with chronic insomnia in the GNHS and GGMP, and used meta-analysis to pool the effect estimate from the GNHS and the GGMP. Meta-analysis of results from the two cohorts showed that the tea consumption was significantly inversely associated with chronic insomnia (Pooled OR: 0.72, 95%CI: 0.55–0.95, $p=0.020$).

We have added the results in the Supplementary Table 11 and in the revised manuscript:

Lines 240-243: *“In addition, meta-analysis of results of the tea consumption-chronic insomnia association from the two cohorts showed that tea consumption (yes versus no) was inversely associated with risk of chronic insomnia (Pooled OR: 0.72, 95%CI: 0.55–0.95, $p=0.020$) (Supplementary Table 11).”*

Lines 746-749: *“In addition, we further investigated the association of tea consumption with risk of chronic insomnia using logistic regression in the GNHS and GGMP, and used random effects meta-analysis to pool the effect estimates from the GNHS and GGMP.”*

15. This manuscript would require significant English language revisions.

Response: We have carefully gone through the manuscript and revised the English language as suggested.

Reviewer #2:

Summary: This research studies whether gut microbiota plays a role in the association between chronic insomnia and cardiometabolic diseases (CMD). Two microbial mediators are identified in a longitudinal cohort and replicated in another independent cohort. Mediator role of bile acids for the association between gut microbiota and CMD is examined using bi-directional mediation analysis. Microbiota-bile acid axis may be a potential intervention target to diminish the impact of chronic insomnia on cardiometabolic health. This manuscript is well organized and well written. The study groups involved are large while one group is used as replication. I have some minor comments as follows.

Response: We thank you for these positive comments.

1. Line 102: why "In the GGMP, participants were divided into two groups: (i) Non-chronic insomnia group, and (ii) Chronic insomnia group (Methods)" while in the GNHS participants were divided into four?

Response: In the GNHS, we have assessed the chronic insomnia status twice: at baseline and at follow-up. At each time point, we can divide them into two groups. Given these longitudinal data, we can characterize the trajectory of chronic insomnia status and divided them into four groups as we stated in our original manuscript (lines 97-103): "(i) Long-term healthy group (i.e., without chronic insomnia at baseline or follow-up), (ii) Recovery group (i.e., from chronic insomnia at baseline to normal at follow-up), (iii) New-onset group (i.e., without chronic insomnia at baseline but with chronic insomnia at follow-up), (iv) Long-term chronic insomnia group (i.e., with chronic insomnia at baseline and follow-up)".

In the GGMP, we assessed the chronic insomnia status only once given its cross-

sectional study design, therefore, we can only divided participants into two groups. We have revised the sentence in the text to clarify the study design. (Lines 103-105)

Lines 103-105: “*In the GGMP, given the cross-sectional study design, participants were divided into two groups: (i) Non-chronic insomnia group, and (ii) Chronic insomnia group (Methods).*”

2. Line 128: Is "Multivariate Analysis by Linear Models (MaAsLin)" the same as a linear regression with multiple predictors? Is chronic insomnia the outcome variable? If so, shouldn't logistic regression be used given chronic insomnia is dichotomous?

Response: In MaAsLin, chronic insomnia is the exposure (predictor) variable while the different microbiota (arcsin-square root transformed abundance) are the outcome variables. Therefore, it is not the same as linear regression with multiple predictors and logistic regression is not appropriate. We have now clarified in the text:

Lines 136-138, “*We used Multivariate Analysis by Linear Models (MaAsLin) to identify potential microbial biomarkers (as outcome variables) of chronic insomnia (as a predictor in the model), adjusted for potential confounders.*”

3. Lines 139-150: What are the results obtained from the "orthogonal partial least squares discrimination analysis (OPLS-DA)"? From which method (OPLS-DA or logistic regression) these p -values are?

Response: We used OPLS-DA method to screen the potential fecal bile acids associated with chronic insomnia. Therefore, the results of OPLS-DA are 10 bile acids, which were presented in the *Extended Data Fig. 5*. Based on the initial findings from the OPLS-DA method, we further used the linear regression model (sorry for the typo in the results section, we have now corrected it. In the method section, we clearly

said that we used the linear regression to do the analysis) to identify the potential bile acids associated with chronic insomnia, adjusted for the potential covariates. The results are presented in the Fig. 2e. Therefore, the p -values are derived from linear regression models.

We have modified the related sentences in the revised manuscript:

Lines 155-158: *“We next used orthogonal partial least squares discrimination analysis (OPLS-DA) to identify potential fecal bile acids associated with chronic insomnia (Extended Data Fig. 5) and then used the linear regression to confirm the chronic insomnia-bile acid associations in the GNHS (Fig. 2e).”*

Lines 662-666: *“Next, we used orthogonal partial least squares discrimination analysis (OPLS-DA) to identify potential bile acids associated with chronic insomnia. For those OPLS-DA selected bile acids, we further used linear regression to confirm the association of chronic insomnia with these bile acids, adjusted for the same covariates as above model 3.”*

4. Line 152: In this section it would be nice to mention the statistical method used, like in the previous sections.

Response: Thank you for the suggestion. As suggested, we have added the related sentences in the revised manuscript:

Lines 175-177: *“To further investigate whether the chronic insomnia-related gut microbiota or bile acids play a role on CMD, we used multivariable logistic regression to examine the association of the chronic insomnia-related gut microbiota or bile acids with CMD.”*

5. Line 215: change "relate" to "related"?

Response: Done (Line 253).

Reviewer #3:

Summary: This is an interesting and important study to examine the associations of insomnia with gut microbiome and its role in cardiometabolic disease risk. Overall, the study is well designed and sufficiently powered, and the analysis appears appropriate. Below are my comments/suggestions to further improve the manuscript.

1. Introduction: the authors need to provide stronger rationales for their study in the introduction. For example, the concept of the brain-gut axis should be mentioned. Also, 1-2 sentences could be added to summarize prior studies showing that gut microbiomes also exhibit circadian rhythms, which interact with the host circadian rhythms, and sleep disturbances such as chronic insomnia could in turn disrupt microbial circadian rhythms, thus influencing gut microbial composition and activity.

Response: We thank you for your suggestions. As suggested, we have revised the manuscript and added more related backgrounds in the introduction section:

Lines 63-68: *“The gut microbiome is vital to the human health^{10,11}. Of note, the brain-gut axis has been intensively studied in the past few years¹²⁻¹⁴. Prior studies have reported that the gut microbiome exhibited circadian rhythms, which showed interaction with the host circadian rhythms¹⁵⁻¹⁷. Sleep disturbances, such as chronic insomnia, could in turn disrupt microbial circadian rhythms, thus influencing gut microbial composition and function¹⁸⁻²².”*

-
2. It seems that the definition and assessment of chronic insomnia were somewhat different between the discovery and validation cohort. What impact do the authors think this difference may have on the results?

Response: It is true that the definition and assessment of chronic insomnia were different, due to the slightly different questionnaires. However, we would like to argue that the difference in the definition is minor and the impact on the results is minimal. In the GNHS, we asked whether they have chronic insomnia for at least 3 days a week for at least six months, while in the GGMP, it was for at least 3 days a week for at least one month. Nevertheless, the best way to test the difference is to compare the results between the two cohorts. Here, our results in the GNHS cohort (as discovery cohort, using a very strict definition of chronic insomnia) were successfully replicated in the GGMP cohort. Thus, although we cannot tell the exact impact of the two definitions on the whole gut microbiome, our results about the chronic insomnia-related gut microbes identified from the GNHS should be reliable. In any way, we have added a sentence in the manuscript to discuss the potential impact of the definition.

Lines 335-338: “Third,, and the potential impact of the two slightly different definitions of chronic insomnia between the GNHS and GGMP is still unclear, although our results are successfully replicated.”

3. Another common way to assess insomnia is to use a symptom score to characterize the severity of the disease. The authors could conduct a secondary analysis evaluating the number of insomnia symptoms in relation to the gut microbial taxa/metabolites identified in the primary analysis. If a dose-response relationship is found, this could provide further support to their study conclusion.

Response: Thank you for the suggestion. We totally agree with this reviewer’s suggestion about the secondary analysis. In the GNHS, we were not able to calculate the chronic insomnia symptom score to conduct the suggested secondary analysis. Nevertheless, in the GGMP, we calculated the chronic insomnia symptom score. As

suggested, we conduct the secondary analysis to evaluate the number of insomnia symptoms in relation to the gut microbial taxa identified in the primary analysis. Multivariable linear regression was used to assess the association of chronic insomnia symptom score with gut microbial biomarkers, adjusted for the same covariates. The secondary analysis results supported our study conclusion.

We have added the results in the Supplementary Table 5 and in the revised manuscript:

Lines 148-153: *“To assess the potential influence of the number of insomnia symptoms, we conducted a secondary analysis in the GGMP, where the data were available, and found that per unit change in the chronic insomnia symptom score was inversely associated with per 1-SD change in Ruminococcaceae UCG-002 (β : -0.04, 95%CI: -0.06--0.02, $p < 0.001$) and Ruminococcaceae UCG-003 (β : -0.04, 95%CI: -0.07--0.01, $p = 0.002$) (Supplementary Table 5).”*

Lines 722-724: *“We conducted a secondary analysis to evaluate the association of the insomnia symptom score (per unit change) with the identified gut microbiota biomarkers by using the linear regression, adjusted for the same covariates.”*

4. The authors claimed that “tea consumption may alleviate the detrimental impact of chronic insomnia on the cardiometabolic health”. However, this suggestion was not directly supported by the results presented. The authors can perform a stratified analysis by tea consumption to examine whether the associations between insomnia-related gut microbiomes and CMD risk factors are weaker among participants with high habitual tea consumption versus those with low consumption.

Response: As suggested, we performed a stratified analysis by tea consumption in the GNHS. The detailed results are shown in the Supplementary Table 10. The results of the inverse association between *Ruminococcaceae UCG-002* and CMD risk factors (especially for diabetes and dyslipidemia) are in general stronger among those with habitual high tea consumption versus low tea consumption groups. These results

support that tea intake may alleviate the detrimental impact of chronic insomnia on the CMD risk via increasing *Ruminococcaceae* UCG-002 (because this microbe is beneficial for CMD and chronic insomnia may decrease it). We added these stratified analyses into the manuscript. Meanwhile, we have thoroughly revised our manuscript to town done our claims about the causality given the observational nature of the cohort study.

Lines 235-240: *“Furthermore, the stratified analysis by tea consumption (yes versus no) in the GNHS showed that the inverse association between Ruminococcaceae UCG-002 and CMD risk factors (especially for T2D (OR: 0.73, 95%CI: 0.60–0.89, p = 0.002) and dyslipidemia (OR: 0.85, 95%CI: 0.74–0.98, p = 0.024)) were in general stronger among those with habitual tea consumption (Supplementary Table 10).”*

Lines 743-746: *“We also performed the additional stratified analyses by tea consumption (yes versus no) using logistic regression in the GNHS to explore whether the associations between chronic insomnia-related gut microbiome and CMD risk factors could be affected by tea consumption.”*

5. For the primary findings, additional stratified analyses (at least by age and sex) are needed to explore potential heterogeneity in the association (it is known that both insomnia and CMD risk are strongly related to age and sex).

Response: Thank you for the suggestion. As suggested, we performed additional interaction analysis by age and sex to explore potential heterogeneity for the chronic insomnia-gut microbiota association and gut microbiota-CMD association in the GNHS and GGMP, and used random effects meta-analysis to pool the effect estimates from the GNHS and the GGMP. We presented the results of stratification analyses in the main text if a significant interaction (potential heterogeneity) was found, and put other stratified results of non-significant interaction in the supplemental tables.

We have added the results in the Supplementary Table 4 and 8 and added the related

sentences in the revised manuscript:

Lines 146-148: *“In addition, chronic insomnia had no interactions with age or sex for Ruminococcaceae UCG-002 and Ruminococcaceae UCG-003 (Supplementary Table 4).”*

Lines 197-202: *“Ruminococcaceae UCG-002 was interacted with sex on the risk of dyslipidaemia ($p_{\text{interaction}} = 0.003$) (Supplementary Table 8). The stratified analyses by sex showed that the inverse association of Ruminococcaceae UCG-002 with dyslipidaemia was significant among men participants (Pooled OR: 0.87, 95%CI: 0.81–0.93, $p < 0.001$), but not among women participants (Pooled OR: 0.96, 95%CI: 0.88–1.05, $p = 0.394$) (Supplementary Table 8).”*

Lines 733-736: *“In addition, we further performed interaction analysis and additional stratified analyses by age and sex to explore potential heterogeneity for the chronic insomnia-gut microbiota association and gut microbiota-CMD association, and used random effects meta-analysis to pool the effect estimates from the GNHS and GGMP.”*

6. Figure 1 can be improved by clearly indicating when stool samples were collected, when insomnia assessment was done and repeated (two time points), and when cardiometabolic risk factors were assessed. Such information can help readers better understand the study design.

Response: Done (Figure 1a).

7. Supplemental Table 2: I would suggest presenting the sample characteristics by chronic insomnia status in the validation cohort too, similar to the discovery cohort. It would be helpful to compare the prevalence of chronic insomnia and understand any potential differences between the two study cohorts.

Response: As suggested, we have now added characteristics by chronic insomnia status in the validation cohort in the Supplemental Table 2.

8. Discussion: The findings on secondary bile acids are very interesting. Some prior studies also support these findings, but the authors did not include them in their discussion. For example, in one study that examined insomnia symptoms and plasma metabolomics (PMID: 30371783), derivatives of second bile acids (e.g., glyoursodeoxycholic acid) were positively associated with more severe insomnia symptoms. Another study found that Ruminococcaceae was related to plasma LDL and triglycerides, which has direct implications for cardiovascular risk (PMID: 31862950).

Response: Thank you for these helpful suggestions. As suggested, we now cited these two important papers and discussed their findings in the discussion section of the revised manuscript as followed:

Lines 285-294: *“The results were consistent with several recent studies^{41,52-55}. One study indicated that secondary bile acid metabolites (i.e., glyoursodeoxycholate) might link poor habitual sleep quality and coronary heart disease risk⁵². Another study showed that Ruminococcaceae UCG-002 was positively associated with insulin sensitivity in patients with polycystic ovary syndrome⁴¹. In addition, another recent study demonstrated that Ruminococcaceae UCG-002 and Ruminococcaceae UCG-003 were positively associated with several plasma HDL subclasses, and were inversely associated with several plasma LDL subclasses, which had direct beneficial implications for cardiovascular health⁵³.”*

9. Discussion: Sleep-disordered breathing often leads to chronic insomnia and poor sleep quality. Intermittent hypoxia from sleep-disordered breathing may also have an impact on gut microbiome, as suggested by some recent evidence (PMID: 33705556, 29896566, 26711739). As sleep-disordered breathing was not considered in the current study, the authors should discuss this and acknowledge this as a limitation (or a potential explanation to their findings)

Response: As suggested, we have added the discussion in the text and acknowledged this as a limitation in the revised manuscript:

Lines 338-341: *“Fourth, we do not collect information on sleep-disordered breathing, which is closely associated with chronic insomnia and may have an impact on gut microbiome and bile acid metabolism⁷⁰⁻⁷².”*

10. Discussion: Given that caffeine in tea may exacerbate insomnia, the authors need to be cautious when interpreting tea consumption as a potential intervention strategy to reduce the adverse cardiometabolic impact of insomnia-related gut microbial alterations.

Response: We agree with you. As suggested, we have revised the related sentences in the discussion to avoid potential confusion caused by our previous causal language (lines 311-313, lines 315-320 and lines 347-348). We also discussed the potential influence of caffeine in tea in the discussion (lines 320-323).

Lines 311-313: *“We found that habitual tea consumption was prospectively associated with the identified gut microbiota and bile acids in an opposite direction compared with chronic insomnia.”*

Lines 315-320: *“Mechanism underlies the association of habitual tea consumption with the gut microbiota-bile acid axis may be attributed to their rich contents of tea polyphenols, flavonoids, alkaloids, and various antioxidant compounds, which are reported to have the ability to modulate the gut microbial composition and bile acid metabolism⁶⁴⁻⁶⁶, and improve the circadian rhythm system presented in the brain and gut^{67,68}.”*

Lines 347-348: *“Habitual tea consumption had an inverse association with the chronic insomnia-disrupted gut microbiota and bile acids.”*

Lines 320-323: *“Nevertheless, we can’t establish a causal relationship between the tea consumption and CMD-related gut microbiota at this stage and these above speculations should be explained with caution, especially given the fact that caffeine in tea may exacerbate insomnia⁶⁹.”*

Reviewer #4:

Summary: The authors investigated the associations between chronic insomnia, bile acid, and CMD. The authors performed a mediation analysis to uncover plausible mechanisms of microbial effects on CMD through specific bile acids. I commend the authors on applying mediation analysis to investigate mechanisms of chronic insomnia on CMD, however their analyses fall short on a few important points mentioned below. Overall, the authors need a more careful consideration of the causal assumptions for mediation analysis and clearer reporting of the methods used.

Response: We thank you for the suggestions about the mediation analysis and causal assumption. Now, we have clearly reported the related results as suggested. The detailed responses to your comments are shown below:

1. It is unclear what the authors mean with the term “bi-directional” mediation analysis. This is not a commonly used term for mediation analysis. Similarly, the term “inverse” mediation is not commonly used.

Response: Thank you for the suggestion. As indicated, bi-directional mediation analysis is not a commonly used word. To avoid potential confusion caused by this word and related “inverse” mediation, we have deleted the term “bi-directional” in the revised manuscript and only keep traditional one-direction mediation analysis (Line

45, Line 211 and Line 692).

We have also revised the sentences in the figure legends:

Figure legends: Lines 867-870: *“The gray lines indicate the associations, with corresponding normalized beta values and p values. The red arrowed lines indicate the microbial effects on CMD mediated by specific bile acid biomarkers, with the corresponding mediation p values. p value < 0.05 is significantly different.”*

2. Most importantly, it appears the authors tried to demonstrate evidence for mediation of microbial effects → bile acid → CMD as opposed to mediation microbial effects → CMD → bile acid. There are two major limitations to this. First, there are six different plausible models of the three variables – the authors only investigated two plausible models. Second, and the most important, comparing the statistical significance and/or magnitude of a mediated effect estimate from one model (e.g., microbial effects → bile acid → CMD) to the statistical significance and/or magnitude of a mediated effect estimate from another model (e.g., microbial effects → CMD → bile acid) does not provide evidence for mediation through one mechanism over the other. These models are from the same equivalence class and therefore cannot be distinguished from one another by statistical tests alone. There must be a compelling scientific reason to suspect that these are the two plausible mediation models and even if there is compelling scientific rationale for these two models, statistics cannot distinguish between these two models. Please see Thoemmes (2015) for more details.

Thoemmes, F. (2015). Reversing arrows in mediation models does not distinguish plausible models. *Basic and Applied Social Psychology*, 37(4), 226–234. <https://doi.org/10.1080/01973533.2015.1049351>

Response: We thank you for these comments, which we found really helpful. We have carefully read the literature you provided on mediation analysis. We again read

related literature about gut microbiota and bile acids (*Thomas, et al, Nature reviews. Drug discovery, 2008, 7(8), 678–693; Schaap, et al. Nature reviews. Gastroenterology & hepatology, 2014, 11(1), 55–67; Song, et al, Microbiome, 2019, 7(1), 9; Pi, et al, mSystems, 2020, 5(3), e00176-20; Taxonomic identification of BSHs in HMP database*). Based on the available evidence, we hypothesized that bile acid biomarkers of chronic insomnia mediated the association of microbial biomarkers of chronic insomnia with cardiometabolic diseases (mediation of microbial effects -> bile acids -> CMD) based on the biological causalities and our findings:

1) In the present study, we demonstrated the trajectory of chronic insomnia status was inversely associated with *Ruminococcaceae UCG-002* and *Ruminococcaceae UCG-003* based on the longitudinal data of chronic insomnia status over the past 6 years. (chronic insomnia -> gut microbiota)

2) In the longitudinal analysis among GNHS, we demonstrated *Ruminococcaceae UCG-002* and *Ruminococcaceae UCG-003* were inversely associated with the cardiometabolic disease risk factor (dyslipidemia). (gut microbiota -> CMD)

3) Gut microbiota, but not humans, have the ability to convert primary bile acids to secondary bile acids (*Witkowski, et al, Circulation research, 2020, 127(4), 553–570*). Previous studies have demonstrated that the family *Ruminococcaceae* harbors BSH and 7 α -dehydroxylase activity, which can convert primary bile acids into secondary bile acids (*Song et al, Microbiome, 2019, 7(1), 9; Pi et al, mSystems, 2020, 5(3), e00176-20; Taxonomic identification of BSHs in HMP database*). *Ruminococcaceae UCG-002* and *Ruminococcaceae UCG-003* are belonged to the family *Ruminococcaceae*, which can convert primary bile acids into secondary bile acids. (gut microbiota -> bile acids)

4) Moreover, *Ruminococcaceae UCG-002* and *Ruminococcaceae UCG-003* in the present study were positively associated with secondary bile acids and were inversely associated with primary bile acids. (gut microbiota -> bile acids)

5) Several studies suggested that treatment with specific microbial derived secondary bile acids (obeticholic acid, deoxycholic acid and glycodeoxycholic acid) in patients with T2D could improve insulin sensitivity and HbA1c (*Mudaliar, et al, Gastroenterology, 2013, 145, 574-582 e571*). (bile acids -> CMD)

6) Previous studies also demonstrated that administration of specific bile acids (e.g. hyocholic acid and hyodeoxycholic acid) in mice improved glucose homeostasis through activation of their receptor Farnesoid-X receptor (FXR) and inhibition of their receptor Takeda-G-protein-receptor-5 (TGR5) (*Makishima, et al, Science, 1999, 284, 1362-1365; Wang, et al, Mol Cell, 1999, 3, 543-553; Kawamata, et al, 2003, J Biol Chem, 278, 9435-9440; Zheng, Cell metabolism, 2021, 33(4), 791-803.e7*). (bile acids -> CMD)

Therefore, based on the above evidence and the hypothesis, we only do the mediation analysis for one model: microbial effects -> bile acid -> CMD.

We have added the sentence in the revised manuscript:

Line 691-694: “*Based on the biological plausibility for the associations among gut microbiota, bile acids and CMD^{54,59,80,81}, and our above findings, we performed the mediation analysis to evaluate whether bile acids could mediate the association of chronic insomnia related-gut microbiota with CMD outcomes (gut microbiota → bile acids → CMD).*”

Line 302-306: “*Ruminococcaceae UCG-002 and Ruminococcaceae UCG-003 may have the ability to convert some primary bile acids into secondary bile acids as they belonged to the bile salt hydrolase (BSH) and 7 α -dehydroxylase-active family Ruminococcaceae, which harbors many secondary bile acid-producing genera such as Faecalibacterium and Ruminniclostridium^{58,59}.*”

3. It is unclear how the authors estimated the mediated effects. There are many statistical methods used in the mediation literature to estimate mediated effects and perform significance testing of mediated effects and it is important that the authors clarify this. Additionally, the authors did not mention the necessary causal assumptions that are required when performing mediation analysis. The interpretation of mediated effects relies on specific no unmeasured confounding

assumptions.

MacKinnon, D. P. (2008). Introduction to statistical mediation analysis. Routledge.

Vanderweele, T. J. (2015). Explanation in Causal Inference: Methods for Mediation and Interaction (Vol. 53). <https://doi.org/10.1017/CBO9781107415324.004>.

Valente, M. J., Rijnhart, J. J. M., Smyth, H. L., Muniz, F. B., & Mackinnon, D. P. (2020). Causal Mediation Programs in R , Mplus , SAS , SPSS , and Stata. Structural Equation Modeling: A Multidisciplinary Journal, 00(00), 1–10. <https://doi.org/10.1080/10705511.2020.1777133>

Response: As suggested, we have now clarified the mediation method in the revised manuscript:

Lines 695-708: *“The mediation analysis was performed to examine the mediating effect of bile acids in the association of chronic insomnia-related gut microbiota with CMD outcomes⁸². We defined three pathways in the mediation analysis: (1) exposure to mediator; (2) mediator to outcome; (3) exposure to outcome. In the mediation analysis, the covariates included: age, sex, BMI, smoking status, alcohol status, physical activity, education, income and total energy intake. The mediation analysis was performed using the R-mediation package with same parameter settings (boot = “TRUE”, boot.ci.type = “perc”, conf.level = 0.95, sims=1000). The total effect was obtained through the sum of a direct effect and a mediated (indirect) effect. Percentage of the mediated effect was calculated using the formula: (mediated effect/total effect) × 100. The sensitivity analysis was performed to test the robustness of the mediation effect and violation of the assumption (sequential ignorability) using R-medsens package with default parameters^{83,84}. The reporting of mediation results followed the Guideline for Reporting Mediation Analyses (AGReMA) statement⁸⁵.”*

As we responded to your second comment, we have a strong biological rationale and hypothesis to do the mediation analysis. We have mentioned the causal assumptions in the revised manuscript:

Introduction section Lines 68-74: *“On the other hand, the gut microbial dysbiosis is associated with the development of CMD, and has a substantial impact on the metabolic health²³⁻²⁹. Meanwhile, the dysregulation of bile acid metabolism and its interaction with gut microbiome are also closely associated with host metabolic health³⁰⁻³³. Repeated sleep disruption in mice has led to a persistent change in gut microbiota composition and changes in bile acid metabolism³⁴⁻³⁶. Therefore, we hypothesized that the gut microbiota-bile acid axis may play a role in linking chronic insomnia and CMD.”*

Method section Lines 691-694: *“Based on the biological plausibility for the associations among gut microbiota, bile acids and CMD^{54,59,80,81}, and our above findings, we performed the mediation analysis to evaluate whether bile acids could mediate the association of chronic insomnia related-gut microbiota with CMD outcomes (gut microbiota → bile acids → CMD).”*

4. There are also sensitivity analyses that are designed to assess the robustness of mediated effects to violations of these assumptions. For example of a sensitivity analysis method see:

Cox, M. G., Kisbu-Sakarya, Y., Miocevic, M., & MacKinnon, D. P. (2013). Sensitivity Plots for Confounder Bias in the Single Mediator Model. *Evaluation Review*, 37(5), 405–431. <https://doi.org/10.1177/0193841X14524576>.

Imai, K., Keele, L., & Yamamoto, T. (2010). Identification, Inference and Sensitivity Analysis for Causal Mediation Effects. *Statistical Science*, 25(1), 51–71. <https://doi.org/10.1214/10-STS321>.

Smith, L. H., & Vanderweele, T. J. (2019). Mediation E-values: Approximate

sensitivity analysis for unmeasured mediator-outcome confounding. *Epidemiology*, 30(6), 835–837. <https://doi.org/10.1097/EDE.0000000000001064>.

Vanderweele, T. J. (2010). Bias formulas for sensitivity analysis for direct and indirect effects. *Epidemiology*, 21(4), 540–551. <https://doi.org/10.1097/EDE.0b013e3181df191c>.

Response: As suggested, we went through the references you provided and then, using the method mentioned in the reference, we performed the sensitivity analysis of mediation with the R-medians package.

We have added the results in the supplementary table 9 and in the revised manuscript:

Lines 221-223: *“Sensitivity analysis for mediation effects indicated that the results of the above mediation analysis were relatively robust to the possible existence of an unmeasured confounder (Supplementary Table 9).”*

Lines 705-707: *“The sensitivity analysis was performed to test the robustness of the mediation effect and violation of the assumption (sequential ignorability) using R-medians package with default parameters^{83,84}.”*

5. The authors would be encouraged to follow the AGRReMA statement for the reporting of mediation results. The following reference contains the necessary reporting guidelines for mediation analysis and many key citations that I would highly encourage the authors consult.

Lee, H., Cashin, A. G., Lamb, S. E., Hopewell, S., Vansteelandt, S., VanderWeele, T. J., ... & Henschke, N. (2021). A Guideline for Reporting Mediation Analyses of Randomized Trials and Observational Studies: The AGRReMA Statement. *JAMA*,

326(11), 1045-1056.

Response: Thank you for the suggestion. As suggested, we reported the mediation results following the AGR_eMA statement and also cited the mentioned reference.

We have added related information in the revised manuscript:

Method section Lines 707-708: “*The reporting of mediation results followed the Guideline for Reporting Mediation Analyses (AGR_eMA) statement⁸⁵.*”

Reviewer comments, second round

Reviewer #1 (Remarks to the Author):

Summary

- This is a much-improved revised version of the manuscript that includes more careful interpretation of the data and appropriate conclusions. It may be beyond the scope of this study, but future animal studies would bolster enthusiasm for the roles of Ruminococcea UCG-002/003 and their associated secondary bile acids (IsoLCA, LCA, UDCA, MCA, and NorCA) in circadian biology and CMD. Minor issues are outlined below.

Minor Issues

- The authors should consider revising the text to exclude the phrase “successfully replicated”. The goal is not to successfully replicate anything, it is simply to test a hypothesis and objectively report the results. Consider revising to, “these results were also observed in the GGMP cohort.”.
 - o Lines 43-45, “Microbial biomarkers of chronic insomnia and their relationships with CMD were successfully replicated in an independent cross-sectional cohort (n=6,122).”.
 - o Lines 143-144, “These results were successfully replicated in the GGMP (Fig.2d).”.
 - o Line 188, “Majority of the results from the GNHS could be replicated in the GGMP.”.
 - o Lines 233-235, “The tea consumption-Ruminococcaceae UCG-association was successfully replicated in the GGMP (β : 0.27, 95%CI: 0.08–0.47; p=0.002) (Fig. 4a).”.
 - o Lines 253-255, “Gut microbial features of chronic insomnia and their relationships with CMD traits were successfully replicated in the independent cohort (GGMP).”
 - o Line 331, “...we replicate our main findings in another large cohort study.”.
 - o Line 338, “...although our results are successfully replicated.”.
- The statistics on Figures 1B and 2B are difficult to interpret because it is so cluttered. Consider using color-coding or the like and organize the p-values more clearly.
- If Figure 2A is representing multivariable linear regression, the data should be presented in a way that shows the representative “fit” of each model. The way it is shown now makes it look like a one-way ANOVA was conducted.
- The manuscript still requires significant English language revisions.

Reviewer #2 (Remarks to the Author):

The authors have adequately addressed my comments. I have no other comments.

Reviewer #3 (Remarks to the Author):

The authors have addressed my prior suggestions, and I have no further comments. Just one note: I don't find main tables in the revised manuscript package (although my comments are not related to the main tables).

Reviewer #4 (Remarks to the Author):

The authors addressed my comments from the previous round. I only have a few remaining comments.

Comments:

I believe the “R-mediation” package that is mentioned on line 702, is actually the R package “mediation” which is described in this paper:

Tingley, D., Yamamoto, T., Hirose, K., Keele, L., & Imai, K. (2014). mediation: R Package for Causal Mediation Analysis. *Journal of Statistical Software*, 59(5), 1–38.

<https://doi.org/10.18637/jss.v059.i05>

I commend the authors for performing a sensitivity analysis. The interpretation the authors provided suggests the estimated mediated effects are relatively robust to unmeasured confounder variables. Because the sensitivity parameter from the `medsens` function in the mediation package is interpreted as a Pearson correlation, the authors may use some guidelines for the interpretation of the magnitude of the correlation. For example, a Pearson correlation of .10 is often regarded as a weak association. This implies that it would take some set of unmeasured confounders of the mediator and outcome to induce a relatively weak correlation between the mediator and the outcome for the observed mediated effect to equal zero.

Because the authors investigated the impact of two types of microbiota on three different bile acids and two different outcomes (Supp. Table 9), this is actually a multiple mediator model. Because the authors estimated these multiple mediator models as separate single mediator models, they are making the strong assumption that the mediators are not correlated with one another conditional on the exposure (microbiota) and the baseline covariates and there is no causal relation between any of the mediators (e.g., MCA does not affect NorCA). This limitation should be discussed in the manuscript. A good reference to understand more about this is listed below:

Imai, K., & Yamamoto, T. (2013). Identification and sensitivity analysis for multiple causal mechanisms: Revisiting evidence from framing experiments. *Political Analysis*, 21(2), 141–171. <https://doi.org/10.1093/pan/mps040>

Response to Reviewers' comments

Reviewer #1:

Summary: This is a much-improved revised version of the manuscript that includes more careful interpretation of the data and appropriate conclusions. It may be beyond the scope of this study, but future animal studies would bolster enthusiasm for the roles of *Ruminococcaceae UCG-002/003* and their associated secondary bile acids (IsoLCA, LCA, UDCA, MCA, and NorCA) in circadian biology and CMD. Minor issues are outlined below.

1. The authors should consider revising the text to exclude the phrase “successfully replicated”. The goal is not to successfully replicate anything, it is simply to test a hypothesis and objectively report the results. Consider revising to, “these results were also observed in the GGMP cohort.”.
 - o Lines 43-45, “Microbial biomarkers of chronic insomnia and their relationships with CMD were successfully replicated in an independent cross-sectional cohort (n=6,122).”.
 - o Lines 143-144, “These results were successfully replicated in the GGMP (Fig.2d).”.
 - o Line 188, “Majority of the results from the GNHS could be replicated in the GGMP.”.
 - o Lines 233-235, “The tea consumption-Ruminococcaceae UCG-association was successfully replicated in the GGMP (β : 0.27, 95%CI: 0.08–0.47; p=0.002) (Fig. 4a).”.
 - o Lines 253-255, “Gut microbial features of chronic insomnia and their relationships with CMD traits were successfully replicated in the independent cohort (GGMP).”
 - o Line 331, “...we replicate our main findings in another large cohort study.”.
 - o Line 338, “...although our results are successfully replicated.”.

Response: As suggested, we have now revised the related sentences in the updated manuscript:

Lines 41-42: *“These results are also observed in an independent cross-sectional cohort (n=6,122).”*

Lines 141-142: *“These results were also observed in the GGMP (Fig.2d).”*

Line 186: *“The majority of the results from the GNHS could be also observed in the GGMP.”*

Lines 231-233: *“The tea consumption-Ruminococcaceae UCG-002 association was also observed in the GGMP (β : 0.27, 95% CI: 0.08–0.47; p=0.002) (Fig. 4a).”*

Lines 250-251: *“The gut microbial features of chronic insomnia and their relationships with CMD traits were also observed in an independent cohort (GGMP).”*

Lines 255-256: “..., and the tea drinking-Ruminococcaceae UCG-002 association was also observed in the GGMP.”

Lines 327-328: “Third, our main findings were also observed in another large cohort study.”

Line 335: “..., although our results were also observed in the GGMP.”

2. The statistics on Figures 1B and 2B are difficult to interpret because it is so cluttered. Consider using color-coding or the like and organize the p-values more clearly.

Response: Done (new Figures 1B and 2B).

3. If Figure 2A is representing multivariable linear regression, the data should be presented in a way that shows the representative “fit” of each model. The way it is shown now makes it look like a one-way ANOVA was conducted.

Response: Done (new Figures 2A).

4. The manuscript still requires significant English language revisions.

Response: As suggested, we have used the Springer Nature Author Services to revise the English language in the revised manuscript.

Reviewer #2:

Summary: The authors have adequately addressed my comments. I have no other comments.

Response: Thank you for your helpful suggestions for improving our manuscript.

Reviewer #3:

Summary: The authors have addressed my prior suggestions, and I have no further comments. Just one note: I don't find main tables in the revised manuscript package (although my comments are not related to the main tables).

Response: Thank you for your helpful suggestions for improving our manuscript. As suggested, we carefully check all tables in the revised manuscript and Supplementary Information again. All the tables are in the supplementary information file.

Reviewer #4:

Summary: The authors addressed my comments from the previous round. I only have

a few remaining comments.

Response: Thank you for your helpful suggestions for improving our manuscript. Please see our responses to your specific comments below:

1. I believe the “R-mediation” package that is mentioned on line 702, is actually the R package “mediation” which is described in this paper:
Tingley, D., Yamamoto, T., Hirose, K., Keele, L., & Imai, K. (2014). mediation: R Package for Causal Mediation Analysis. *Journal of Statistical Software*, 59(5), 1–38. <https://doi.org/10.18637/jss.v059.i05>

Response: As suggested, we have revised our statement about the package name to R package “mediation” (Line 523).

2. I commend the authors for performing a sensitivity analysis. The interpretation the authors provided suggests the estimated mediated effects are relatively robust to unmeasured confounder variables. Because the sensitivity parameter from the medsens function in the mediation package is interpreted as a Pearson correlation, the authors may use some guidelines for the interpretation of the magnitude of the correlation. For example, a Pearson correlation of .10 is often regarded as a weak association. This implies that it would take some set of unmeasured confounders of the mediator and outcome to induce a relatively weak correlation between the mediator and the outcome for the observed mediated effect to equal zero.

Response: As suggested, we have added the guidelines for the interpretation of correlation magnitude into the footnotes of Supplementary Table 7.

3. Because the authors investigated the impact of two types of microbiota on three different bile acids and two different outcomes (Supp. Table 9), this is actually a multiple mediator model. Because the authors estimated these multiple mediator models as separate single mediator models, they are making the strong assumption that the mediators are not correlated with one another conditional on the exposure (microbiota) and the baseline covariates and there is no causal relation between any of the mediators (e.g., MCA does not affect NorCA). This limitation should be discussed in the manuscript. A good reference to understand more about this is listed below: Imai, K., & Yamamoto, T. (2013). Identification and sensitivity analysis for multiple causal mechanisms: Revisiting evidence from framing experiments. *Political Analysis*, 21(2), 141–71. <https://doi.org/10.1093/pan/mps040>

Response: As suggested, we have added the discussion in the text and acknowledged this as a limitation in the revised manuscript:

Lines 338-341: *“Fifth, we conducted the mediation analysis for multiple bile acids using separate single mediator models; however, it is possible that these bile acids are highly correlated with each other or even have a causal association with each other, which needs further investigation.”*